# Temperate forests can deliver future wood demand and climate-change mitigation dependent on afforestation and circularity

Eilidh J. Forster [1] ✉, David Styles[1,2] & John R. Healey [1] ✉

Global wood demand is expected to rise but supply capacity is questioned due to limited forest resources. Additionally, the global warming potential (GWP) impact of increased wood supply and use is not well understood. We propose a framework combining forest carbon modelling and dynamic consequential life-cycle assessment to evaluate this impact. Applying it to generic temperate forest, we show that afforestation to double productive forest area combined with enhanced productivity can meet lower-bound wood demand projections from 2058. Temperate forestry value-chains can achieve cumulative GWP benefit of up to 265 Tg $CO_2$-equivalent ($CO_2e$) by 2100 per 100,000 ha of forest (if expanded to 200,000 ha through afforestation). Net GWP balance depends on which overseas forests supply domestic shortfalls, how wood is used, and the rate of industrial decarbonisation. Increased wood-use could aid climate-change mitigation, providing it is coupled with a long-term planting strategy, enhanced forest productivity and efficient wood use.

Global wood demand has been rising by 1.1% per annum over the last 20 years[1] and is projected to grow at even higher rates[2–4], due to population growth, economic growth[4] and the transition towards a net zero bioeconomy[5,6] as per the Paris Agreement[7]. However, high levels of uncertainty surrounding wood demand projections[5], and key knowledge gaps on the impact of wood demand on terrestrial carbon flux[8] and lifecycle greenhouse gas emissions[9–11] remain.

Integrated assessment models (IAM) are often used in global analyses of the environmental impact of major economic activities, to assist informed policy-making in the context of climate change[12–14]. However, IAM typically fail to account for the importance of region-specific management of existing forests and investment in managed forest, which could lead to underestimating forest carbon flows[8]. In contrast, forest land-sector focussed economic models (known as forest sector models, FSM[15,16]), such as Global Timber Model (GTM)[17], Global Biosphere Management Model (GLOBIOM)[18], and Global Forest Products Model (GFPM)[19], can incorporate heterogeneity in the forest resource base, as well as ecological constraints, management opportunities, product markets, and land use and management responses to market and environmental change[15,20]. FSM are typically used to

project global wood demand under different economic and climate policy scenarios[3,5,6]. However, important local variations in tree species composition, growth rates and forest management practices within regions may not be captured[15], and these models rely on uncertain assumptions about future environmental, macroeconomic and specific forest market conditions[21]. Therefore, substantial uncertainty remains about biophysical supply meeting projected demand[8], and with competing demands on land use there is a risk of 'over-stated reliance' on wood resources in net-zero decarbonisation plans[22].

Supplementing IAM with further analytical approaches is recommended[23] to increase resolution of analysis[24,25] and to reality-check scenarios[22]. Combining IAM and high-resolution forest carbon modelling, Blattert et al.[22] found disparity between IAM-projected demand for wood needed for achieving EU net zero greenhouse gas (GHG) emissions targets and available supply calculated using national-scale modelling. This disparity is an under-considered policy conundrum[22] and it raises the question addressed in the present study - what lifecycle global warming potential (GWP) consequences could result from increased wood demand: (i) being supplied via combinations of augmented (national) temperate forest value chains and

[1]School of Environmental and Natural Sciences, Bangor University, Bangor, Gwynedd, UK. [2]School of Biological & Chemical Sciences and Ryan Institute, University of Galway, Galway, Ireland. ✉e-mail: eilidhforster@gmail.com; j.healey@bangor.ac.uk

overseas imports, and/or (ii) facing supply shortfalls, prolonging dependency on non-wood product alternatives via supply chain displacement?

Addressing this question requires a finer resolution of analysis than is possible using IAMs and is beyond the scope of FSM and forest growth models. Lifecycle assessment (LCA) offers a rigorous methodology[26] for analysing these carbon-critical[9,27] value chain impacts. There have been recent examples of LCA being integrated with IAMs for modelling low carbon power systems[28] and improving prospective product assessment[3,29]. LCA has also been integrated with forest growth modelling[30,31] to account for the GWP impact of expanded forestry value chains. These are advanced by the framework developed in the present study (Supplementary Methods 1 and Supplementary Fig. 1), which builds on detailed prospective dynamic LCA modelling of entire forest-wood value chains developed and explained in Forster et al.[9,27]. Crucially, the full downstream greenhouse gas (GHG) mitigation consequences of wood use and end-of-life management are considered, including carbon storage and material and energy substitution within a decarbonising future economy. Novel aspects include calculating potential future deficit in wood-supply by comparing projected wood-demand curves to wood-supply from a range of modelled, augmented, and expanded national/regional forest management scenarios; and linking the supply deficit to marginal expansion of supply from other regions.

In order to illustrate application of our enhanced framework and provide insights that have broad relevance, we analyse shifts in management strategy and expansion (afforestation) for a generic temperate forest under two wood demand projections - 'low' and 'high', defined as linear growth rates of 1.1% and 2.3% per annum respectively, from 2023, and equating to a demand increase of 30% and 62% by 2050, respectively (see Methods below). We assess a range of forest management strategies through which a country's existing forest and land resources could supply increased wood demand, including increasing the rate of production from existing forest resources, and expanding the area of productive forest.

As a reference scenario we use a 100,000 ha Sitka spruce forest managed on a clear-fell 50-year rotation (from planting to harvest) - in which there are 2000 ha of forest in each year age-class. We model eight forest management and expansion scenario combinations, defined in Tables 1 and 2. The management scenarios include (1) 'reference' rotation; (2) intensification of 'reference' production through a 10% 'shortened rotation' to 45 years, phased over a 50 year period (i.e. 10,000 ha are shortened to 45 years every 5 years to avoid 'shocks' to the ecosystem and to annual wood supply); (3) extensification of 'reference' production through a 10% 'extended rotation' to 55 years, over 50 years; and (4) replanting after 'reference' harvest with 'higher productivity' (25% increase in yield), faster-growing trees. Sensitivity of results to two scales of natural disturbance was tested by modelling 'lower productivity', caused by a ramp up to 15% or 30% of 'reference' forest area experiencing reduced carbon sequestration and production output, proxied through a yield decline of 33% for one harvest rotation across affected areas (defined in Table 1, and results presented in Supplementary Data 3).

The expansion scenarios increase the area of each of these four (existing forest) management scenarios through afforestation, applying the same rotation length and productivity criteria (i.e. for the 'shortened rotation' scenario the new forest is harvested from year 45; and for the 'higher productivity' scenario the new forest is planted with trees of that higher productivity from year 1). Expansion is modelled at two different afforestation rates: 2000 ha/yr for 50 years, equating to a doubling of forest area; and 1000 ha/yr for 50 years equating to a 50% increase in forest area. Both afforestation rates are within current policies for low forest cover nations[32–34]. To test the sensitivity of the results to the duration of forest expansion (afforestation period) we also calculate the impact of shortening the duration from 50 to 35

years. These hypothetical land-use scenarios help reveal relationships between forestry value chains and climate mitigation that are relevant to temperate forests at the regional or country level whilst not being tied to the unique situation of a specific country (e.g. the scale, productivity and age classes of its forest resources).

We used the Carbon Budget Model for the Canadian Forestry Sector (CBM-CFS3)[35] to model terrestrial carbon dynamics (forest carbon storage and emissions) and LCA to quantify GWP impact (measured in kg $CO_2$e) from forestry operations through wood processing and HWP-manufacturing to (cascading) product use(s), over a 100-year study period ($GWP_{100}$, referred to as GWP from now on). We account for upstream and direct processing emissions, HWP carbon storage, avoided emissions from product substitution, land use (and management) change, and indirect land use change (caused by a change in imported HWP). More details are provided in the Methods below.

## Results

### Supply-demand deficit

We find that in a temperate country context for the augmented forest management scenarios tested there is a large gap between realistic biophysical potential wood supply and a rate of increased wood demand at the high end of the range projected in previous studies (Fig. 1). For this 100,000 ha hypothetical temperate forest, a doubling of area in combination with modified forest management is insufficient to consistently meet even the low end of projected demand increases. In all modelled scenarios, there will be increased reliance on imports (indicated in red on Fig. 1) over the next 35–55 years, for both 'low' and 'high' wood demand projections. This is because changes to rotation length have minimal impact on wood supply – illustrated by the small shift in existing forest production rates from the 'reference' scenario line (Fig. 1b, c, f, g), and contributions from afforestation or replanting with 'higher productivity' forest only materialise when trees reach harvest age. 'Lower productivity' in existing forest due to natural disturbances (ramping up over 15 years to affect 30% of existing forest area) could reduce annual wood production by up to 32% (relative to 'reference' existing forest wood production across 100,000 ha) for a 15-year period and further increase reliance on imports (Supplementary Data 3).

'Higher productivity' existing forest supplies 21% more wood compared to 'reference' and is the only scenario in which already existing forest (vs new forest), shown in dark blue (Fig. 1), provides a contribution to marginal wood demand (i.e. the difference between 'projected demand' and 'reference' lines) on average over the study period. With existing forest and new forest combined, 'higher productivity' supplies in aggregate 57% and 93% more wood than 'reference' (baseline, no expansion) for 'low-' and 'high-expansion', respectively. In contrast, retaining baseline productivity, the combined existing and new 'reference-rotation' forest supplies in aggregate, 25% and 51% more wood than 'reference' (baseline, no expansion) for 'low-' and 'high-expansion', respectively.

The 'high expansion' scenarios could produce sufficient wood to exceed 'low' projected demand increase in the longer term (from year 35-55 onwards, scenario dependent, indicated in pink, Fig. 1e–h). However, if projected wood demand increase is 'high', no scenarios will supply this (Fig. 1).

A supply deficit and reliance on imports is exacerbated if the duration of afforestation is shortened from 50 to 35 years, which leads to pronounced dips in future supply (indicated by the boundary between the pale and dark grey segments, Fig. 1). Committing to a minimum afforestation duration equal to the forest rotation length would help avoid major dips in future supply.

Next, we analyse the GWP impact of supplying projected wood demand via the domestic and import supply balances for each of the forest management scenarios portrayed in Fig. 1.

**Table 1 | Forest management and expansion scenario options modelled in the study**

| Scenario description | Scenario name | Yield Class (YC) | Harvest | Rotation length | Afforestation rate | Afforestation duration |
|---|---|---|---|---|---|---|
| **(a) Existing forest** | | | | | | |
| 'Reference' | Rotation_50_50 | 18 | Clear fell, with thinning in year 21 | 50 years | n/a | n/a |
| Shortened rotation | Rotation_50_45 | 18 | Clear fell, with thinning in year 21 | Shift from 50 years to 45 years (gradual over 50-year period) | n/a | n/a |
| Extended rotation | Rotation_50_55 | 18 | Clear fell, with thinning in year 21 | Shift from 50 years to 55 years (gradual over 50-year period) | n/a | n/a |
| Higher productivity | Rotation_50_35 | 18, 24 after replanting | Clear fell, with thinning in year 21 (YC 18) or year 18 (YC24) | 50 years then 35 years after replanting | n/a | n/a |
| Lower productivity | 15% YC12 or | 18, 12 after first harvest on 15% of forest area, 18 after second harvest | Clear fell, with thinning in year 21 | 50 years | n/a | n/a |
| Lower productivity | 30% YC12 | 18, 12 after first harvest on 30% of forest area, 18 after second harvest | Clear fell, with thinning in year 21 | 50 years | n/a | n/a |
| **(b) Forest expansion** | | | | | | |
| 'Reference' rotation – with expansion options | Expansion_50 | 18 | Clear fell, thinning in year 21 | 50 years | 1000 ha/yr or 2000 ha/yr | 35 years or 50 years |
| Shortened rotation – with expansion options | Expansion_45 | 18 | Clear fell, thinning in year 21 | 45 years | 1000 ha/yr or 2000 ha/yr | 35 years or 50 years |
| Extended rotation – with expansion options | Expansion_55 | 18 | Clear fell, thinning in year 21 | 55 years | 1000 ha/yr or 2000 ha/yr | 35 years or 50 years |
| Higher productivity – with expansion options | Expansion_35 | 24 | Clear fell, thinning in year 18 | 35 years | 1000 ha/yr or 2000 ha/yr | 35 years or 50 years |
| Broadleaf – with expansion options | Expansion_BL | 6 | Unharvested - mixed broadleaf, (BL) | n/a | 1000 ha/yr | 35 years or 50 years |

Part (a) defines the existing forest management options modelled in the study using the forest growth model, CBM-CFS3. Part (b) defines forest expansion options modelled in the study using the forest growth model. 'Higher productivity' refers to replanting with faster-growing trees for the next rotation after harvested at 50 years (then managed on a 35-year rotation thereafter). 'Lower productivity' refers to the replanted trees on a proportion of the area only achieving lower growth rates (yield) for one rotation, to represent reduced production due to, e.g., pests, pathogens, wind, fire and drought. The two degrees of 'lower productivity' are based on a 33% yield reduction on either 15% or 30% of forest area, respectively. Results for 'lower productivity' scenarios are presented in Supplementary Information. 'Forest expansion' (afforestation) is modelled with options of afforestation rate and period.

**Table 2 | Forest management and expansion scenario combinations modelled in the study**

| Scenario description | Existing forest | Forest expansion: Management | Forest expansion: Afforestation duration | Forest expansion: Afforestation rate | Wood demand projection |
|---|---|---|---|---|---|
| Reference | Rotation_50_50 | n/a | n/a | n/a | Low or High |
| Reference-rotation, high expansion | Rotation_50_50 | Expansion_50 | 50 years | 1000 ha/yr* or 2000 ha/yr | Low or High |
| Reference-rotation, low expansion** | Rotation_50_50 | Expansion_50 | 35 years | 1000 ha/yr* or 2000 ha/yr | Low or High |
| Shortened-rotation, high expansion | Rotation_50_45 | Expansion_45 | 50 years | 1000 ha/yr* or 2000 ha/yr | Low or High |
| Shortened-rotation, low expansion** | Rotation_50_45 | Expansion_45 | 35 years | 1000 ha/yr* or 2000 ha/yr | Low or High |
| Extended-rotation, high expansion | Rotation_50_55 | Expansion_55 | 50 years | 1000 ha/yr* or 2000 ha/yr | Low or High |
| Extended-rotation, low expansion** | Rotation_50_55 | Expansion_55 | 35 years | 1000 ha/yr* or 2000 ha/yr | Low or High |
| Higher productivity, high expansion | Rotation_50_35 | Expansion_35 | 50 years | 1000 ha/yr* or 2000 ha/yr | Low or High |
| Higher productivity, low expansion** | Rotation_50_35 | Expansion_35 | 35 years | 1000 ha/yr* or 2000 ha/yr | Low or High |
| Lower productivity, high expansion | 15% YC12 | Expansion_50 | 50 years | 1000 ha/yr* or 2000 ha/yr | Low or High |
| Lower productivity, low expansion** | 15% YC12 | Expansion_50 | 35 years | 1000 ha/yr* or 2000 ha/yr | Low or High |
| Lower productivity, high expansion | 30% YC12 | Expansion_50 | 50 years | 1000 ha/yr* or 2000 ha/yr | Low or High |
| Lower productivity, low expansion** | 30% YC12 | Expansion_50 | 35 years | 1000 ha/yr* or 2000 ha/yr | Low or High |

Includes the reference forest (with no afforestation), and the four (including reference) temperate forest management scenarios with afforestation options, under both high and low timber demand projection scenarios. * to maintain clarity of presentation, the scenarios of 'Expansion_BL', i.e. 1000 ha/yr of mixed broadleaf afforestation (See Table 1) are excluded from this table. ** low expansion scenarios also present net results for 50-year afforestation to show the comparative impact of shortening the duration of the afforestation period.

## GWP impact under low increase in wood demand

Without modified forest management or expansion ('reference', red line, Fig. 2), cumulative net GWP impact will remain positive and increase steadily over time (i.e. increasing wood demand will drive an increase in cumulative net GHG emissions). This is because static domestic wood production leads to rising annual imports (in line with demand), resulting in overseas forest C storage losses that exceed GWP "credits" derived from carbon (C) storage and product substitution by overseas forest HWP for scenarios modelled in this study (see below, 'Relative impacts of marginal imports'). In other words, terrestrial C losses from forest degradation caused by increases in HWP imports are not fully compensated by the associated HWP C storage and product substitution benefits.

'Low expansion' can deliver cumulative GWP mitigation of -174 Tg $CO_2e$ over 100 years with the 'higher productivity' scenario (Fig. 2d). Only 'higher productivity' achieves mitigation via enhanced overseas forest C storage (i.e. terrestrial C gain, blue bars, Fig. 2d, h) – a result of domestic supply exceeding projected demand on average. The varied rotation scenarios (Fig. 2a–c) provide much lower cumulative GWP mitigation of −84, −102 and −104 Tg $CO_2e$ for 'shortened-rotation', 'reference-rotation' and 'extended-rotation', respectively. 'Lower productivity' in (an area ramping up to) 30% of existing forest for one rotation (and reducing annual wood production by 32% relative to 'reference' existing forest wood production across 100,000 ha, for a 15 year period) would reduce cumulative GWP mitigation by 12% to -90 Tg $CO_2e$ ('lower productivity net impact', pink line, Fig. 2a). Furthermore, annual (i.e. increasing cumulative) GWP mitigation is only sustained for the whole study period with a 50-year afforestation duration. Shortening the afforestation duration to 35 years for 'low expansion' scenarios reduces cumulative mitigation by 27–46% (equating to 39-46 Tg $CO_2e$), scenario dependent, and for all scenarios except 'higher productivity' the value-chain even becomes a net $CO_2e$ emitter during the final three years of the study period (indicated by the rising trajectory of cumulative net GWP impact from year 2119, dotted line, Fig. 2a–c).

'High expansion' scenarios deliver 59-73% more cumulative GWP mitigation than 'low expansion' (Fig. 2e–h) due to the impact of greater 'new forest' C sequestration and 'new forest HWP' GWP benefits, combined with lower dependency on imports. 'Increased productivity' (Fig. 2h) again delivers the most notable cumulative GWP mitigation, at −277 Tg $CO_2e$ over 100 years (54–104% more than the varied rotation scenarios). Cumulative GWP mitigation delivered by the varied rotation scenarios remains relatively close when expansion is 'high', at −136, −177 & −180 Tg $CO_2e$ for 'shortened-rotation', 'reference-rotation' and 'extended-rotation', respectively. Shortening the afforestation duration from 50 to 35 years for 'high expansion' scenarios reduces cumulative GWP mitigation by 25–36% (equating to 34–70 Tg $CO_2e$), scenario dependent. Although net GWP mitigation is sustained during the study period, it diminishes under the shortened afforestation duration scenarios, as indicated by the levelling out of cumulative net GWP mitigation (dashed lines, Fig. 2e-h).

If projected demand is 'low,' all management scenarios with 'high expansion' can support net cumulative GWP mitigation, and this mitigation can be doubled by 'increased productivity'. Varied rotation management scenarios have more limited effects, confirming that expansion of productive forestry and yield enhancement are the most important forest-related strategies to maximise climate change mitigation.

## GWP impact under high increase in wood demand

When projected wood demand is 'high', results become dominated by the impacts of marginal imports (Fig. 3, blue and red bars), which are

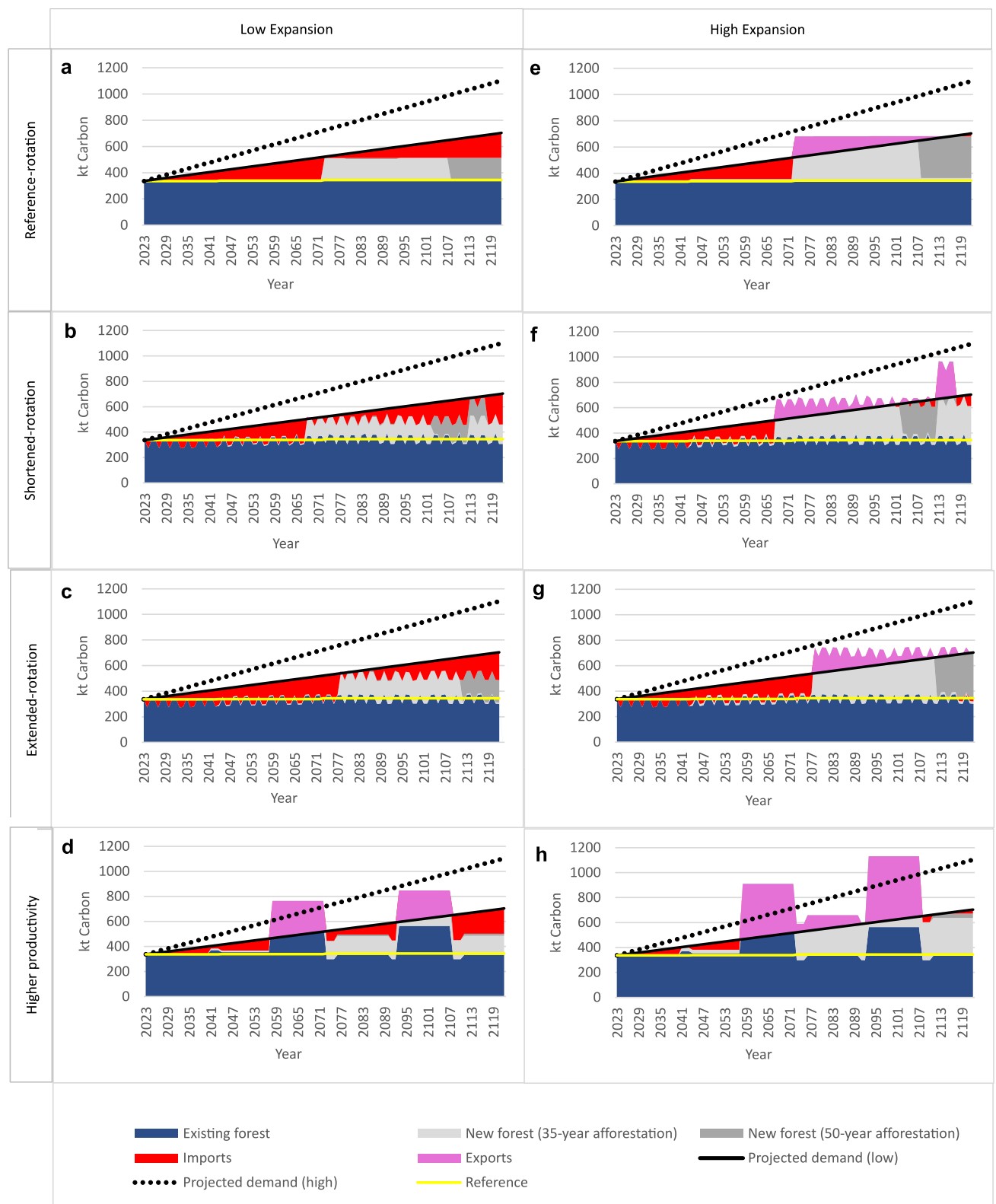

highly dependent on the balance between overseas forest C storage (CO₂e losses) and C storage and production substitution "credits" attributable to 'overseas forest HWP'. Despite this, all but one of the modelled scenarios results in cumulative net GWP mitigation by the end of the study period (Fig. 3a, c–h). Only 'shortened-rotation, low expansion' (Fig. 3b) leads to cumulative net GWP impact exceeding zero in the final few years of the study period. However, across every

scenario, cumulative GWP mitigation at the end of the study period is 70–95 Tg CO₂e less than in the respective 'low projected demand' scenario (Fig. 2). Additionally, although cumulative GWP mitigation is sustained for all but one scenario, the value-chain becomes a net CO₂e emitter from 2107 onwards in the varied rotation, low expansion scenarios, as indicated by the upward trajectory of net GWP impact (blue line).

**Fig. 1 | Annual wood supply and demand for eight management and expansion scenarios for an existing 100,000 ha of temperate forest.** In the reference year, domestic supply ('existing forest') is in balance with demand. In all forest management scenarios we balance changes in projected demand with changes in supply – by modifying imports (or exports) to make up any deficit (or surplus) in domestic (temperate forest) wood production. These scenarios comprise existing forest under four different management applications (1. 'reference-rotation' (**a**, **e**), 2. 'shortened-rotation' (**b**, **f**), 3. 'extended-rotation' (**c**, **g**) and 4. 'higher productivity' (**d**, **h**)). Detailed definitions are provided in Methods. Each existing forest management application is paired with two different expansion rates – 'low' (**a**–**d**, 1000 ha/yr) and 'high' (**e**–**h**, 2000 ha/yr). In addition, the new forest is represented as two afforestation durations – 35 years (pale grey) and 50 years (years 36–50 shown in dark grey). Production from existing forest and new forest are plotted against a 'reference' scenario (business as usual management of existing forest, i.e. reference-rotation with no expansion or increase in productivity, yellow line) and two wood demand projections: 'projected demand (low)' (solid black line) and 'projected demand (high)' (dotted black line), representing 1.1% and 2.3% linear annual growth rates, respectively. The difference between 'projected demand (low)' and the combined existing plus new forest (with 50-year afforestation duration) supply, is calculated as 'imports' (red) or 'exports' (pink), depending on whether supply is lower than or exceeds (respectively) the 'projected demand (low)' curve. The 15-year 'peaks' in temperate production observed for 'higher productivity' scenarios (**d**, **h**) result from a shift in rotation from 50 years to 35 years (exacerbated by a 50-year afforestation duration for a 35-year rotation forest), leading to 15-year periods when (existing forest and new forest) harvested area doubles. 'Projected demand (high)' is indicated by the black dotted line but demand-supply balances for 'projected demand (high)' are not illustrated here.

## Relative impacts of marginal imports

The range of possible non-temperate forest sources that are likely to be needed to meet shortfall in domestic wood supply in a temperate country versus projected increased future demand have potentially substantial, though very different, GWP implications. If shortfall is met by shortening the rotation of overseas old-growth boreal forest from 128 years to 68 years ('Boreal 1'), $CO_2e$ emissions would exceed the GWP impact of the alternative, prolonged use of non-wood product and fuel alternatives ('curtailed HWP supply') when accounting for optimistic, progressive industrial decarbonisation during the study period (Fig. 4). If shortfall is met by harvesting boreal stumps and residues ('Boreal 2 & 3'), continuation of conventional logging in tropical forest ('Tropical CVL') or by 'Tropical (afforestation)', associated $CO_2e$ emissions would not exceed the GWP impact of 'curtailed HWP supply'.

Of the scenarios considered in this study, the existing forest source for increasing imported wood supply with the lowest GWP-impact is extraction of stumps and harvesting residues in boreal systems that are already managed at moderate harvest intensity (68-year rotation), at 0.17 Mg $CO_2e$ per $m^3$ wood (Fig. 4). However, given the limited range of HWPs this lower quality wood is suited to, additional wood sources would also be required. Reducing harvest rotation in old-growth boreal forest (i.e. from 128 years to 68 years) results in the highest GWP per $m^3$ wood, at 1.85 Mg $CO_2e$ per $m^3$. In contrast, increasing supply from tropical afforestation can offer net C sequestration benefits, of −0.9 Mg $CO_2e$ per $m^3$ wood. When tropical afforestation is included in calculation of the 'average' impact of sources of marginal wood imports, the GWP impact drops from 0.78 (the value used to calculate 'marginal import/export forest' for modelled scenarios in Figs. 2 and 3) to 0.45 Mg $CO_2e$ (indicated by the dotted black line, Fig. 4). Notably, the value switches from being higher than 'avoided HWP-use' (i.e. the impact of continuing to use non-HWP alternatives) to lower. This would result in marginal imports deriving a net GWP "credit" per $m^3$, instead of a net GWP burden per $m^3$ as considered in Figs. 2 & 3. These results illustrate the particular sensitivity of the net balance between (average) net forest $CO_2e$ emissions and HWP GWP "credits" associated with increased harvest demand to a range of counterfactual assumptions across entire value chains (full life cycle perspective). Thus, small changes in (counterfactual) assumptions may propagate to large net changes in net balance, and thus fundamentally alter conclusions of studies evaluating forest management and harvest strategies[3]. Therefore, studies with unclear or truncated boundaries could easily derive misleading results through systematic bias in the calculation of this balance.

## Discussion

Results of this study provide important insights on relationships between climate mitigation, wood supply, demand and use that have broad relevance for temperate countries that intend to use and/or supply more wood; and they highlight the importance of expanding boundaries beyond the focal country – a key component of the proposed framework (Supplementary Method 1, Supplementary Fig. 1). However, given sensitivity of results to forest productivity, afforestation rate, wood demand and potential impact on overseas forests, tailoring key scenario parameters (Supplementary Table 1) to represent specific geographical contexts is recommended if conclusions drawn are intended to inform national policy or action.

Although options for increasing wood supply before 2050 in most temperate countries are clearly limited, action started now and sustained to increase forest area and productivity could close the long-term supply-demand gap within the lower range of demand projections. However, the upper range of future demand projections cannot be met by temperate forests, implying a possible need to moderate prospective wood demand and use as the bioeconomy expands[36]. A focus on reducing process losses[5], and maximising value in cascading and circular value chains[37] could regulate demand[9,36] whilst also achieving considerable HWP C storage and product substitution credits[6,9].

There is an important need to relate bioeconomy (industrial) expansion back to feedstock availability to avoid unrealistic expectation of supply, especially for forestry, owing to the time needed to grow trees. Therefore, integrated assessment and modelling used in the context of climate change mitigation should consider forestry (land use) and wood value chains as a policy package: critically, linking increased harvesting to forest expansion and improved productivity to security of future wood supply and enhanced decarbonisation. Increased wood demand is known to stimulate productive forest expansion[38] due to market influences and timber price elasticity[39]. However, if expansion is inhibited, e.g. via absent supporting policy or restrictive land use policy, thereby limiting supply, then harvest leakage is very likely to occur[17,20,40], i.e. a compensatory increase in harvesting elsewhere. Ignoring market influences and the response of supply to demand in climate change mitigation studies[3] could lead to wood- and land-use conclusions that hinder net-zero efforts. Restricting wood-use would delay bioeconomy growth and global decarbonisation[41], yet increasing wood-use without expanding forest resources and enhancing productivity could lead to net $CO_2e$ emissions.

When low projected demand is supplied via augmented (national) temperate forest value chains and overseas imports this is expected to lead to net GWP benefits. In all those scenarios modelled here, net cumulative $CO_2e$ emissions remain below zero. However, when projected demand is high, net annual $CO_2e$ emissions ultimately occur after 2107. Therefore, limiting wood demand through improved value-chain efficiencies is also an important strategy to prevent undermining net climate-change mitigation benefits of future wood use, whilst also sustaining growth of the bioeconomy[9,36]. It is clear that increased wood use is not, in itself, a climate-change solution[42], unless afforestation, increasing forest productivity under sustainable forest management, and demand moderation through enhanced circularity and cascading of wood use are also integrated into the strategy[9,36].

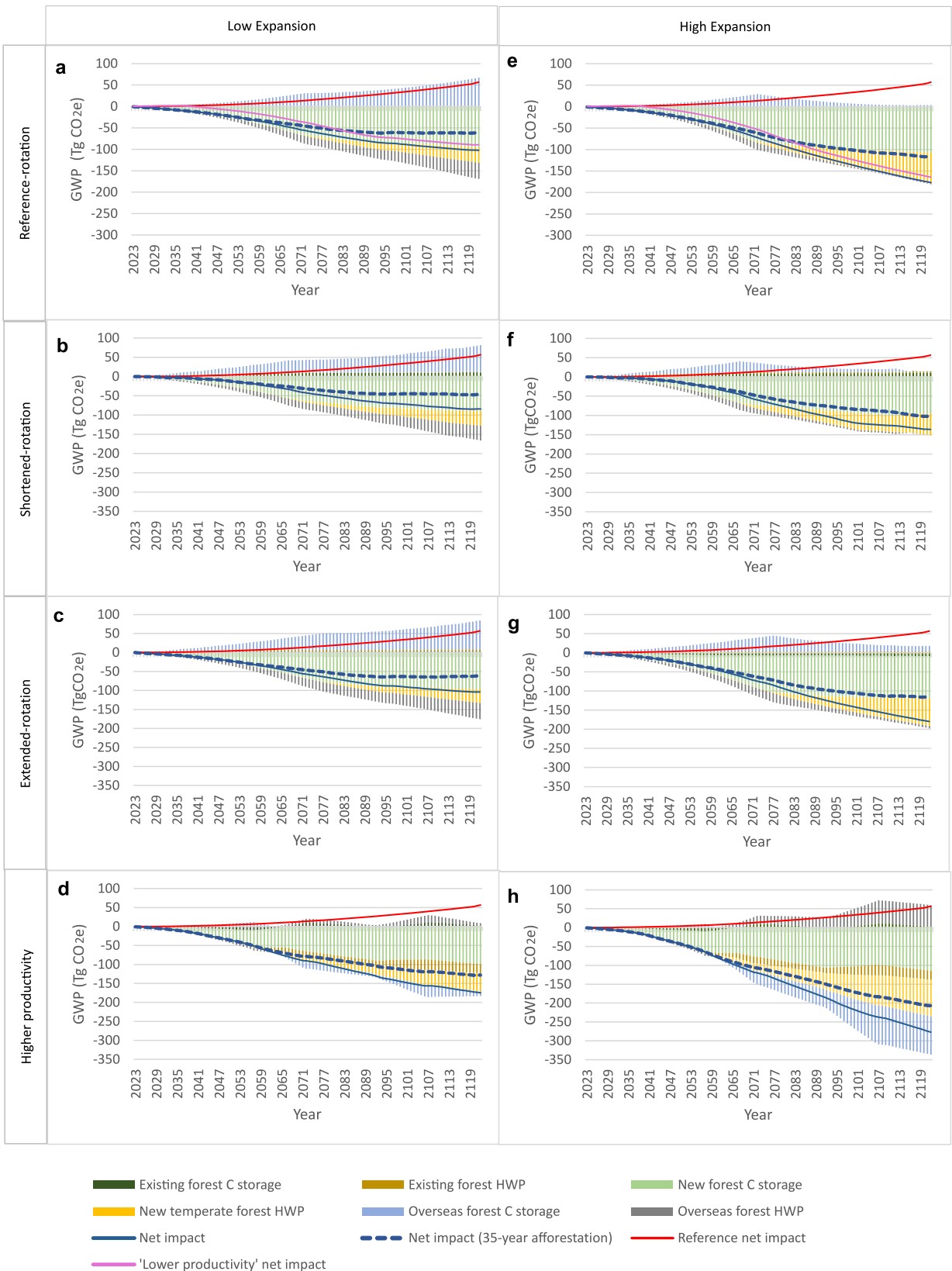

The relative GWP impact of prolonging dependency on non-wood product alternatives via supply chain displacement versus supplying increased wood demand depends on the source of wood. Reliance on wood imports to deliver increasing domestic demand and territorial net-zero targets risks undermining global (overseas) climate-change mitigation efforts. The range of GWP burdens associated with wood imports to temperate countries considered in this study provides a conservative bounding of impact. The net GWP balance between overseas forest C storage loss and GWP credits associated with HWP use is very sensitive, and can tip from net mitigation to a net $CO_2e$ source (Fig. 4), depending on where the wood is sourced from (the age, management history and growth rate

**Fig. 2 | Cumulative marginal GWP effects, relative to a 2023 baseline, for eight management and expansion scenarios of a 100,000 ha temperate forest under 'low' projected demand.** These scenarios comprise existing forest under four different management applications (1. 'reference-rotation' (**a**, **e**), 2. 'shortened-rotation' (**b**, **f**), 3. 'extended-rotation' (**c**, **g**) and 4. 'higher productivity' (**d**, **h**)). Each existing forest management application is paired with two different expansion rates – 'low' (**a–d**, 1000 ha/yr) and 'high' (**e–h**, 2000 ha/yr). Scenarios are defined in detail in Tables 1 and 2. The scenarios are plotted against a reference-rotation with no expansion or productivity increase ('business as usual' – red line). 'Low' and 'high expansion' represent 1% and 2% linear temperate forest expansion per year, respectively. All results for new forest shown here assume a 50-year afforestation duration, unless otherwise indicated. As a sensitivity analysis the 'net impact (35-year afforestation)' is shown (blue dashed line), illustrating the GWP effect of shortening afforestation to 35 years. Additionally, 'lower productivity net impact' (pink line, panels **a**, **e**) shows the effect of reducing productivity by one-third in a forest area ramping up to 30% of the 'reference-rotation' forest, for one rotation

(scenario description in Table 1, full results in Supplementary Information). All results represent marginal changes in carbon (C) storage and product substitution "credits" from the 2023 baseline for the focal generic temperate country. 'Existing forest C storage' indicates change in terrestrial C storage (i.e. net forest $CO_2$ flux) from altered management of the existing 100,000 ha 'reference' forest (shortened-rotation, lengthened-rotation or increased productivity). 'Overseas forest C storage' reflects changes in terrestrial C storage from increased/decreased harvesting in non-temperate forests in other countries due to supply adjustments (imports or exports) required to meet projected demand – described further in Methods. 'Existing forest HWP' shows changes in HWP C storage plus product substitution 'avoided emissions' from changes to the management of the 'reference' forest. 'New forest HWP' represents the change in HWP C storage plus product substitution arising from newly planted temperate forests. 'Overseas forest HWP' refers to changes in HWP C storage plus product substitution related to changes in imported/exported harvested wood products needed to balance supply with demand in the focal country.

of forests) and how it is used. Given this sensitive balance and the dominant impact it has on overall GWP results, it is imperative that forestry studies evaluating climate mitigation should transparently consider a comprehensive range of land use (change) scenarios as well as full downstream wood use consequences – as enabled by the framework developed in the present study. Notably, IAM do not include cascading uses of wood[23,43], underestimating whole value-chain (life cycle) GWP mitigation and possibly implying net emission increases for some uses, which could steer inferred climate action away from additional wood use.

There is potential to expand wood supply in tropical areas in a manner that increases terrestrial C storage[44–47], which therefore implies GWP mitigation associated with imports from these areas. However, in a world where all countries are likely to experience increasing demand for wood simultaneously[5], and given an absence of control over land management in other countries[48–50], relying on this possibility would be naive, and comes with numerous socio-economic[51] and biodiversity conservation[20] caveats. The average GWP burden attributed to wood imports in this study is conservative, supporting robust conclusions.

There are important product substitution GWP benefits of using wood (e.g. in construction), and C stored in HWP can provide a C sink[27,52]. In addition to fossil fuel substitution benefits, bioenergy with carbon capture and storage (BECCS) also has promise for further enhancing the C sink[53–55] by providing a route to permanent geological storage of biogenic carbon. Based on conservative assumptions around GWP impacts of imported wood and GWP credits from future wood use, the results presented here demonstrate potential for considerable, yet constrained, expansion of wood-use in the bioeconomies of temperate countries to drive overall climate-change mitigation. To achieve this potential, national net-zero policies must connect climate-change mitigation in wood-using sectors with wood supply (expansion) in the land sector, and prospective demand may need to be moderated via support for cascading and circular use of wood[9]. There is an urgent need for more integrated evidence that incorporates holistic assessment of prospective forestry value chains alongside landscape dynamics (including forest management and expansion), at both national and global scales, including improved estimates of the potential impacts of climate change-linked threats to the future productivity of both temperate forests and the other forest biomes providing wood.

## Methods
### Scope of Life Cycle Assessment
The scope of the LCA is represented in Fig. 5, showing the main factors considered within the modelling boundaries. A more detailed process diagram is included in Supplementary Information.

We account for terrestrial (soil and biomass) carbon (C) storage, harvested wood product (HWP) C storage, substitution of materials and fossil fuels, and long-term sequestration of biogenic C via future deployment of Bioenergy with carbon capture and storage (BECCS), over a 100-year period (using the same assumptions as Forster et al.[27] and summarised here). Expanded LCA boundaries (Fig. 5) encompassed: (i) forest management change (due to shifts in rotation and/or productivity); (ii) land use change due to temperate afforestation on spared agricultural grassland; (iii) forest establishment; (iv) forest growth; (v) forestry operations; (vi) debarking; (vii) sawmilling (including drying, planing and chemical treatment); (viii) wood panel production; (ix) paper and paperboard production; (x) bioenergy generation, including BECCS; (xi) credits for avoided use of fossil fuels (substituted energy generation and mineral construction material production – see Supplementary Table 1 and Supplementary Fig. 2 for further detail); (xii) C storage in HWPs related to 'decay' (retiral) functions[37], and (xiii) recycling and disposal of retired HWPs, including via (x). The production and transport of all material and energy inputs were accounted for, as were the construction or manufacture of infrastructure and capital equipment. Full life cycle inventories are provided in Supplementary Data 1, with an example table for the Hierarchical wood-use value chain in Supplementary Table 2. Material flows were derived using data from a combination of forest C modelling[35], forest product-breakout data (at forest gate) from over 2,000 ha of commercially managed forests, data from a commercial sawmill that maximises sawn-wood output[56], UK national wood recycling data[57] and timber-use statistics[58] – elaborated in Supplementary Data 1. These assumptions are considered representative of temperate forestry value chains more broadly than the UK. Commercial forest management practices (e.g. clear-fell harvesting) and sawmill technology are similar across regions. Given the focus of this paper on GHG mitigation, only the global warming potential (IPCC 2013 GWP100a[59]) impact category was evaluated, expressed as kg carbon dioxide equivalent ($CO_2$e) emissions.

We assess the GWP impact of the following forest management, wood demand and supply variations for a temperate country, relative to business-as-usual management of existing forest in the baseline year of 2023:
- Changes to existing forest management and expansion of forest area
- Increased or reduced rotation length in existing forest
- Increased or decreased growth rate of existing forest
- Afforestation – temperate forest expansion (at a fixed rate of 2000 ha per year), with a varying proportion of (i) new commercial (conifer) forest subsequently harvested for wood

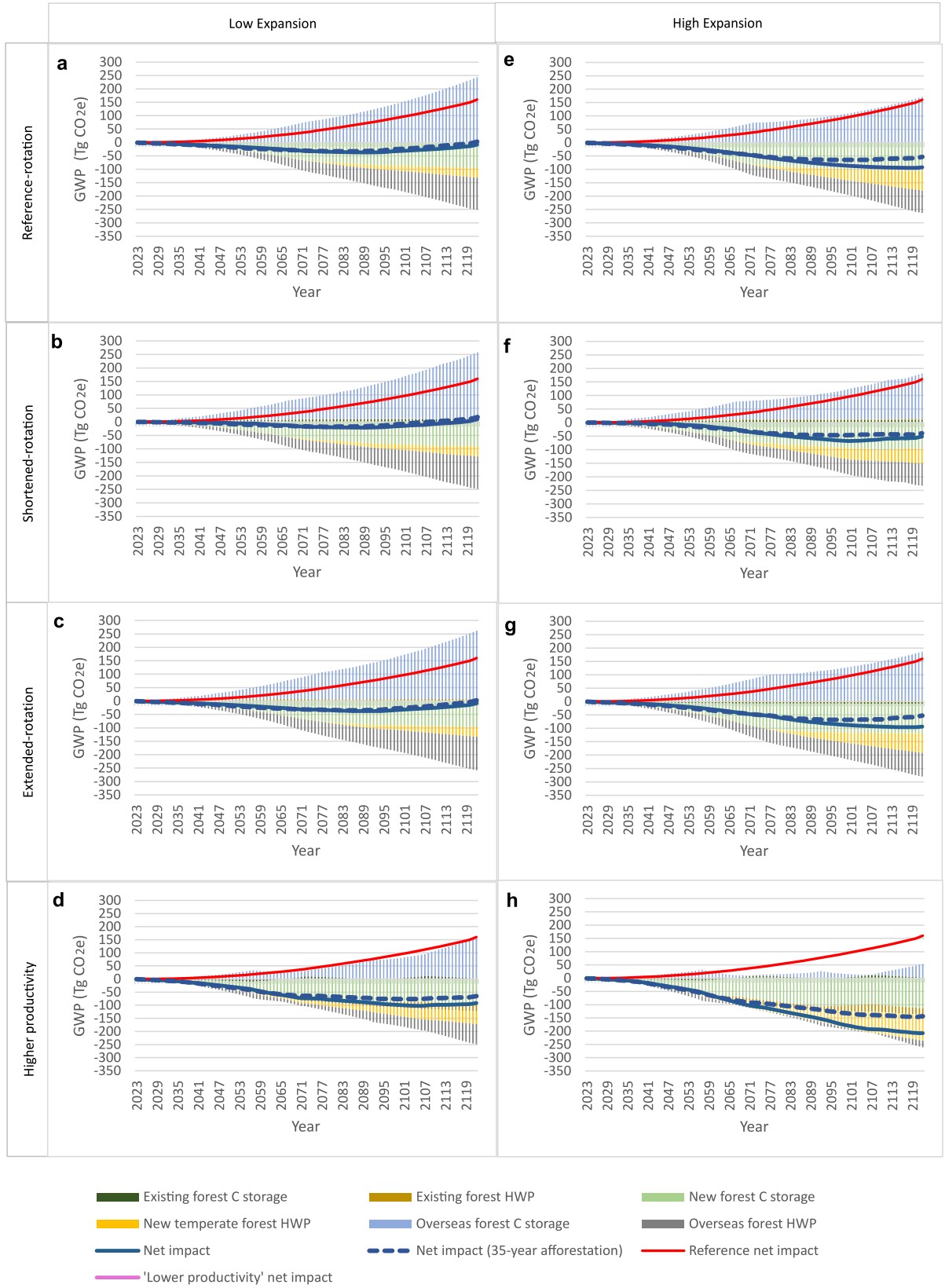

production relative to (ii) new forest of broadleaved tree species characteristic of semi-natural forests, left unharvested throughout the study period. Management of new commercial forest is the same as existing forest

- An increase in imports required to meet marginal demand shortfall in the within-country wood production relative to annual demand, through different options to intensify or expand production in various forest types in other exporting countries:

**Fig. 3 | Cumulative marginal GWP effects, relative to a 2023 baseline, of eight management and expansion scenarios for an existing 100,000 ha of temperate forest under 'high' projected demand.** These scenarios comprise existing forest under four different management applications 1. 'reference-rotation' (**a**, **e**), 2. 'shortened-rotation' (**b**, **f**), 3. 'extended-rotation' (**c**, **g**) and 4. 'higher productivity' (**d**, **h**)). Each existing forest management application is paired with two different expansion rates – 'low' (**a**–**d**, 1000 ha/yr) and 'high' (**e**–**h**, 2000 ha/yr). Scenarios are defined in detail in Tables 1 and 2. The scenarios are plotted against a reference-rotation with no expansion or increase in productivity (i.e. continuation of business as usual – red line). 'Low' and 'high expansion' mean temperate forest expands linearly by 1% and 2% per year, respectively. All results relating to new forest shown in Fig. 3 are for an afforestation duration of 50 years, unless otherwise indicated. As a sensitivity analysis 'net impact (35-year afforestation)' is calculated and plotted to show the GWP impact of shortening the afforestation duration to 35 years (blue dashed line), for each scenario. The results of additional sensitivity analysis exploring forest disturbance events on cumulative GWP effects are presented in SI and Supplementary Data 3. All results

relate to marginal changes in carbon (C) storage and product substitution "credits" from the 2023 baseline for the focal country in which the temperate forest is located. 'Existing forest C storage' relates to change in terrestrial C storage (i.e. net forest $CO_2$ flux) caused by shifting management of the existing 100,000 ha 'reference' forest (to shortened-rotation, lengthened-rotation or increased productivity). 'Overseas forest C storage' is the change in terrestrial C storage in non-temperate forests in other countries due to increased or decreased harvesting (i.e. increased imports or exports to the temperate country) in order to meet projected demand – described further in Methods. 'Existing forest HWP' is the change in C storage plus product substitution 'avoided emissions' associated with increased/decreased harvesting arising from shifting management of the existing 100,000 ha 'reference' forest (to shortened-rotation, lengthened-rotation, or increased productivity). 'New forest HWP' is the change in C storage plus product substitution arising from the harvest of new temperate forest planted nationally. 'Overseas forest HWP' is the change in C storage plus product substitution associated with the change in imported/exported harvested wood products required to balance supply with demand in the focal country.

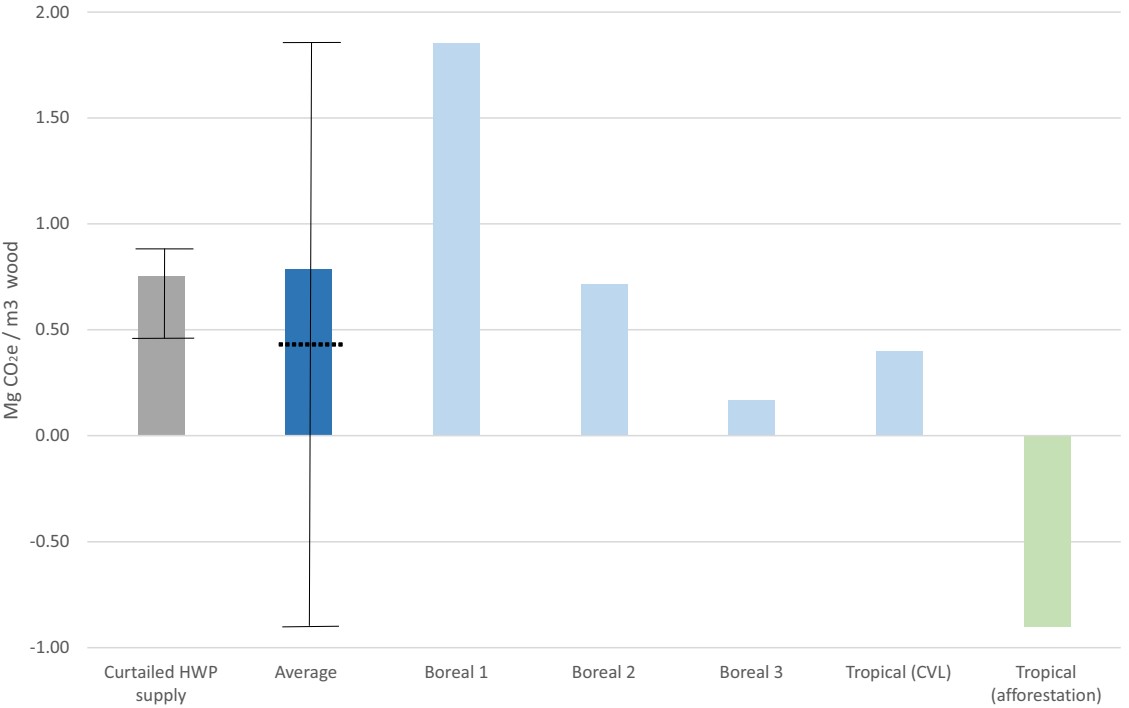

**Fig. 4 | 'Overseas forest C storage' GWP impact expressed per m³ of marginal imported wood supply.** Five forestry scenarios are considered: 'Boreal (1)' is shortening rotation of old-growth boreal forest from 128 years to 68 years; 'boreal (2)' is shortening rotation from 128 years to 68 years and harvesting stumps and residues; and 'boreal (3)' is harvesting stumps and residues (from a boreal forest already managed on a 68-year rotation). 'Tropical (CVL)' is the $CO_2$e cost of continuing conventional logging instead of shifting to reduced impact logging in tropical moist forest. 'Average' is the average (net forest $CO_2$e flux) of the preceding four scenarios – this is used to calculate 'overseas forest C storage' impact

in Figs. 2 & 3. The black dotted line shows the shift in 'average' if the 'tropical (afforestation)' scenario is also included in the average calculation. The solid black lines on 'Average' show the range in GWP impact of all five forestry scenarios in this figure. 'Curtailed HWP supply' is the time-average GWP impact per m³ of marginal wood not supplied (i.e., the $CO_2$ equivalent emissions due to consumption of non-wood product substitutes, in this case, concrete and fossil fuels). The solid black lines on 'curtailed HWP supply' indicate the range in GWP over the model period as a result of progressive industrial decarbonisation (i.e., reducing over time). Details of modelled scenarios are in Methods.

- Old-growth boreal forest (shortening rotation only)
- Old-growth boreal forest (shortening rotation, with thinning and removal of tree harvest residues)
- Tropical forest (continuation of more intensive conventional logging of natural forest rather than shifting to reduced impact logging)
- Afforestation with new plantations in a tropical country
- Continued reliance on fossil fuel-derived materials – if marginal wood demand is not supplied.

Full definitions of the scenarios modelled are provided in Tables 1 and 2.

## Forest growth model

Forest C modelling is performed using the Carbon Budget Model for the Canadian Forestry Sector CBM-CFS3[35]. Note we do not model forest albedo effects on global warming nor the effects of climate change (i.e. warming) on forest growth. Whilst these are important factors that affect the climate change mitigation efficacy of forests[60] their net effects remain uncertain and would apply similarly across the study scenarios so would not be expected to alter the study findings. There is increasing evidence of the major impact of disturbances due to pests, pathogens, wind, fire and drought, linked to global warming impacts, on forest productivity[60–66]. While these future risks are highly uncertain and would apply similarly across the main study scenarios

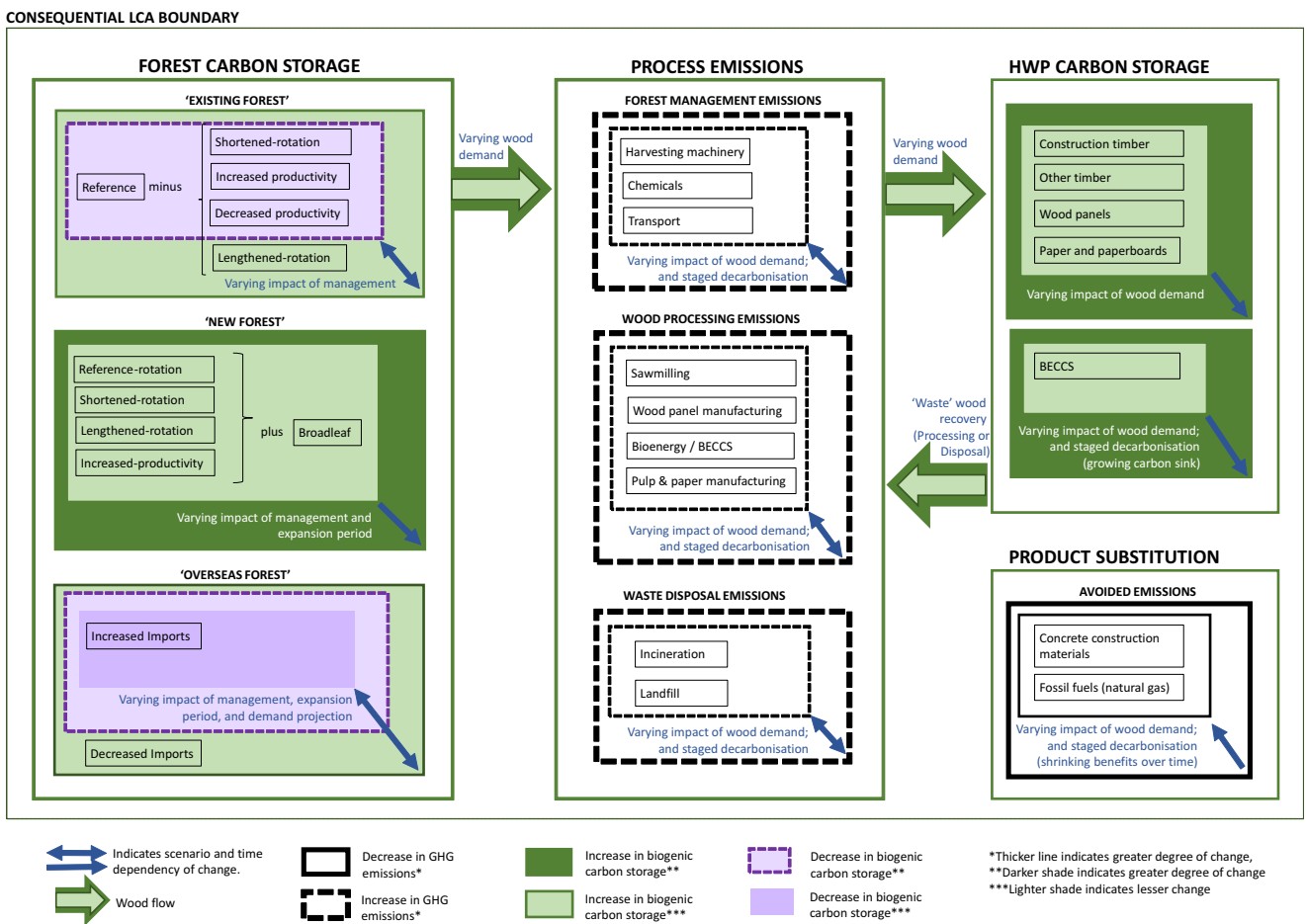

**Fig. 5 | Consequential life cycle assessment system boundary.** As well as illustrating the scope of the consequential LCA by the components included within the boundary line, this diagram also indicates the scenario and time dependency of changes over the 100-yr model period. Analyses are run every year for the scenarios to include annual changes in forest carbon stocks and harvested wood flows, and decadal changes in avoided emissions factors and BECCS deployment over the 100-yr model period. The direction of change through time for dynamic impacts is indicated by the blue arrows, which are scenario and time dependent. Depending on forest management and expansion strategy and wood demand, forest carbon storage could increase (green) or decrease (purple). Increasing wood demand increases HWP carbon storage (green) but leads to increasing process emissions (hatched black lines); whilst avoided emissions from product substitution (solid thick black lines) shrink over time due to decadal decarbonisation. We calculate the global warming potential (GWP) impact of forestry value chain (system) changes from year-0 baseline 'business as usual' (BAU) due to changes in: (i) 'existing forest', temperate forest management; (ii) 'new forest' areas of temperate afforestation; (iii) 'overseas forest' management, in order to meet projected wood demand increases. Changes in 'existing forest' refer to forest management in existing temperate forests in the focal country. 'New forest' refers to afforestation in the focal country. 'Overseas' land-use change refers to the changes in intensity of harvests from boreal and tropical forests assumed to make up marginal demand shortfalls from temperate forests. BECCS is bioenergy with carbon capture and storage. See Fig. 1 and Table 1 for further details of modelled scenarios. A more comprehensive LCA boundary diagram indicating all the main process steps in the model is provided in Supplementary Fig. 2, which together with Supplementary Table 2 includes additional detail on HWP substitution assumptions.

we have included two additional scenarios (described below) to assess how their potential negative impacts on forest growth would affect harvest volumes and GWP mitigation trends.

Further key methodology assumptions are described below.

### Forest management
The study 'reference' is a 100,000 ha 'normal' forest (i.e. comprising an even distribution of annual age classes) of Sitka spruce, yield class (YC) 18, managed with a 50-year clear-fell rotation, with thinning in year 21 (all representative of current forest management of temperate oceanic environments such as those in UK and other countries within its region). All alternative management scenarios modelled start with these 'reference' conditions in year 0 of the study period, which is 2023.

The study explores the GWP impact of altering forest rotation length and tree growth rate, and expanding forest area. We quantify the GWP impact of modifying forest management (from the reference) in combination with options for expanding forest area. The full range of management and expansion (afforestation) options that are assessed in this study are presented in Tables 1 and 2.

### Altering rotation length
Since the 'reference' forest is already managed close to the optimal (commercial) rotation length for typical temperate plantations, large shifts are unlikely. Therefore, we model shortening and extending rotation length by 10% from the reference scenario, implementing this shift gradually over a 50-year period (i.e. 10% of the forest area is transitioned every 5 years) so as to limit the rate of change in annual harvest volumes (which would be constrained by wood value chain and market capacity) and forest C dynamics.

### Increasing productivity
Harvested trees (at 50 years) are replanted with YC24 Sitka spruce, managed on a 35-year rotation. This equates to a 25% increase in

productivity such as could be possible with an enhanced breeding programme[67], from 17.2 to 21.5 m³/ha/yr.

## Decreasing productivity

Harvested trees (at 50 years) are replanted with Sitka spruce, managed on a 50-year rotation. Ramping up over 15 years, 15% or 30% of forest area regrows at YC12 (the rest remains at YC18) for one 50-yr forest rotation, then returns to YC18 when replanted after the next harvest. This equates to reduced wood production of 16% or 32%, respectively (across the whole existing forest area, relative to 'reference' existing forest), for a 15 year period. These two scenarios represent two intensities of production losses that could arise due to, e.g., pests, pathogens, wind, fire and drought. These are intended as a sensitivity test and the full results of these scenarios are provided in Supplementary Data 3.

## Afforestation

There is a physical limit to the land available for expansion of forest area through afforestation in temperate countries, and its implementation is further constrained by multiple social, economic and political factors: the timber market, jurisdictional regulation and policy[36].

We assume that commercial plantation afforestation is part of a comprehensive strategy to achieve fixed targets for increase in total forest area (as is the case for national policies in many temperate countries, such as the UK[32,33]). However, such policies in temperate countries typically do not specify the types of forest planted. While commercial factors and the demand for strong land-based carbon sinks to achieve 'net zero' favour fast-growing conifers[27], biodiversity conservation and delivery of other cultural and regulatory ecosystem services may favour establishment of unharvested forests comprising broadleaved tree species typical of semi-natural forests. Therefore, we calculate the consequential impact of varying the proportion (high and low) of commercial conifer to broadleaved species in the afforestation strategy. The 'high' afforestation strategy equates to a doubling over 50 years of the area of commercial conifer plantation (with wood harvested at the end of the rotation) and the 'low' strategy equates to a 50% increase in commercial conifer plantation area over 50 years, with the balance of the total afforestation area comprising broadleaved species. In relation to the reference forest area, this translates to commercial conifer plantation afforestation rates of 2000 ha/yr and 1000 ha/yr, respectively. In the 'low' strategy we calculate the impact of the marginal increase (1000 ha/yr) in broadleaf forest area comprising a mixture of sycamore, silver birch, oak and rowan, with an average growth rate of YC6, that is unharvested during the study period, a typical scenario for the low-productivity land most readily available for large-scale afforestation (in the UK).

To assess the impact of afforestation duration at these planting rates, on both wood supply and GWP impact, we also model a shorter afforestation duration of 35 years for each afforestation strategy, applying the same annual planting rates of 2000 ha/yr and 1000 ha/yr (resulting in a 30% smaller area of new forest planted than for the standard 50-year afforestation strategies). The full range of management and expansion (afforestation) options that are assessed in this study are presented in Tables 1 and 2.

## Projected wood demand

We use the FAO definition of industrial round wood (IRW)[68], which is all round wood except wood fuel. It includes sawlogs, veneer logs, pulpwood and other IRW and, in the case of trade, chips and particles, and wood residues. The majority of IRW production is traded in the form of HWPs[69], i.e. IRW that has already been processed (normally in the production country), such as sawnwood, wood-based panels, and paper and paperboard. Therefore, in the present study, we use the term 'wood' to inclusively refer to IRW and/or HWP, unless differentiation is important for clarity.

The Low demand increase projection is based on historic global wood production rates[69], which are similar to rates in Europe, both at 1.1% average linear growth per annum over the last 20 years. We assume that future demand growth continues at this linear rate. However, we decouple a second High demand increase projection from historic trends to account for growth of the bioeconomy, including projected demands reported in published IAM and FSM studies, as described below.

Projections of increased global IRW demand by FAO (using The Global Forest Products Model) range between 27% and 44% for 2020-2050, depending on efficiency of residue use (70% and 30%, respectively); and a further increase of up to 14% is possible if trends for timber construction and man-made cellulosic fibres (MMCF) in textile production increase[5], equating to a possible 58% rise globally between 2020 and 2050, or 1.9% per year on average. There is also large regional variation in projected changes in demand[2–5,22], with the greatest increases predicted in Eastern Asia, which is expected to further increase its consumption up to 41% of the world´s primary processed wood products by 2050. This is an increase in demand of 56% between 2020 and 2050, equating to linear annual demand growth of 1.9% during 2020-2050. Coupled with the additional potential 14% increase in global demand linked to trends in timber construction and MMCF, the projected average increase in annual demand is 2.3% per annum over this period. Other studies project wider variations still. Demand in East Asia and the Pacific has been projected to rise by between 2%[2] and 4.4% per annum to 2050[3] for sawnwood and wood panels combined (total IRW demand, not separately reported, can be interpreted to rise at a similar rate since demand for paper and paperboard – the other major traded HWP group – has similar growth projections). Meanwhile, Europe and Central Asia have relatively lower projected sawnwood demand increases of 0.5%[2] to 0.6%[3] per annum to 2050. National-scale projections of demand increase for Finland, Sweden, Norway and the temperate region of Germany (Bavaria), range from 34% (Sweden) to 40% (Norway) between 2020 and 2050 (interpreting growth curves)[22]. Notably, these countries are important timber production regions, together contributing 29% of Europe's IRW production[70]; and they already have high wood consumption per capita, relative to (low production) countries such as the UK, Ireland and the Netherlands[70], so the potential for percentage demand increase may be tempered by this.

There is, however, great uncertainty surrounding projected demand for IRW because of its dependence on social, political, economic and environmental systems that are 'non-stationary', with correlations between variables changing over time[71]. It is therefore unsurprising that the previous studies reviewed above have reported a wide range of predicted increases in future demand for IRW and typically limit their predictions to 30- to 40-year timeframes, rarely projecting far beyond 2050[2–5,22]. This means that the extended projection of published growth rates to 2122, applied in the high-demand scenario in the present study, is unavoidably very uncertain.

To account for the wide range of published projections and their great uncertainty, we selected two contrasting scenarios for the rate of increase in future demand for IRW to represent the lower and higher estimates in the range of previous studies, respectively. This is important to assess the sensitivity of net value chain GWP impact to this variation in future demand. We selected annual linear growth of 1.1% to represent a 'low' demand projection scenario. It matches historic growth in wood consumption globally and in Europe, according to FAOSTAT[69]. It also closely matches historic growth in the UK, a temperate developed country with moderate per capita wood consumption[70], over that last 20 years. It equates to a 30% demand increase by 2050, or an 85% increase by 2100. For the 'high' demand projection we use 2.3% annual growth, which equates to a 62% demand

increase by 2050, or a 177% increase by 2100, which is the highest case regional scenario (for Eastern Asia) derived from FAO modelling. However, at a country level, demand increase could potentially be even higher due to further variation within regions.

## Harvested wood products impact

We assume a 'hierarchical' value chain breakout for wood flows that remains constant throughout the study period. The assumptions and methodology for calculating the GWP impact of processing and use of HWP under this hierarchical value chain are taken from Forster et al.[27] (Supplementary Data 1), including the decarbonisation projections and product substitution effects. The current study advances the work by Forster et al.[27] by calculating the GWP impact of dynamic annual harvests from both existing and new forest. A HWP GWP impact calculation module (Supplementary Data 2) was developed (from Supplementary Data 1) for this purpose and used in the present study.

The same GWP impact calculation methodology is applied to all HWP, including marginal imports and exports. The volume of marginal imports (or exports) is calculated as the difference between projected demand and supply from commercial plantations within the temperate country (illustrated in Fig. 1).

## Marginal imports forest impact

We assume that marginal imports (i.e. import differences vis-a-vis the 2023 baseline year) are supplied from non-temperate forests in order to gauge the range and scale of potential consequential impact if temperate regions cannot increase production sufficiently to meet their own projected demand. For this we have treated boreal forests as a different category from temperate forests, which is particularly appropriate as a major component of wood demand in temperate countries is softwood from conifer trees, and boreal forests represent a major source of this softwood in global trade[5].

This calculation has high uncertainty for multiple reasons:

- Future product breakout (wood use) is uncertain, i.e. how wood will be used in the future due to developments in technology and the bioeconomy[37]
- Product breakout from different forest types and regions varies greatly, due to variation in productivity, wood properties and quality, and local uses
- There is a wide range of options for increasing harvest volumes across regions (e.g., shortening rotations[72] or increasing productivity[73])
- There is great complexity and challenge in accurately modelling C fluxes in different forest management scenarios, and in many cases, there are limited available data.

We therefore take a simplified approach to modelling the GWP impact from fluxes in forest C stocks ('GWP (forest C)') of marginal imports, with the intention of estimating the potential scale of impact, rather than attempting a precise dynamic representation of impact. We calculate the GWP (forest C) impact of the following four selected forest management change scenarios in tonnes of $CO_2$e per marginal $m^3$ of harvested wood. Boreal scenarios (1, 2 & 3) account for changes in above- and below-ground C. The tropical scenario accounts for change in above-ground C only.

Boreal (1) – boreal, managed on a 128-year rotation with no thinning changed to a 68-year rotation with no thinning and no removal of harvesting residues[72])

Boreal (2) – boreal, managed on a 128-year rotation with no thinning changed to a 68-year rotation with moderate thinning and removal of harvesting stumps and residues for processing[72])

Boreal (3) – boreal, managed on a 68-year rotation with moderate thinning changed by introduction of removal of harvesting stumps and residues[72])

Tropical – continuation of more intensive conventional logging practice in natural tropical forest, instead of changing to low impact logging[74]. This scenario will maintain a higher rate of supply of tropical hardwood, which has limited potential to substitute for softwood in major markets (assuming historic wood-use trends) but is included in our study as an "outgroup" comparator for the scenarios of intensification of boreal forest harvesting.

To avoid complex temporal C dynamics caused by transitioning from one forest management regime to another, we assume an instant shift from steady state in the initial system to steady state in the new (higher supply) system. We calculate the change in average C stocks and average annual volume of harvested wood for each scenario. Whilst this doesn't capture important temporal C dynamics, it indicates the potential scale of impact of sourcing marginal IRW from different forest resources and highlights areas where further study is important. See Supplementary Data 3 for calculations.

We relate the change in average C stocks to each additional $m^3$ of wood harvested, for each of the four scenarios above (Boreal 1, 2 & 3 and Tropical), and then calculate the 'Average' forest impact of each of these scenarios (their sum, divided by four) – this average value is used to calculate 'Marginal import/export forest' in Figs. 2 and 3. We scale up the 'Average' GWP impact of 'Marginal imports/exports forest' from tonnes of C (as $CO_2$e) per-$m^3$ of marginal harvested wood to the calculated $m^3$ of marginal imports/exports for each modelled scenario to quantify their respective GWP impacts.

Note that we apply the same product breakout assumptions used for the temperate plantations to these four different sources of imported wood, i.e. we do not account for variation in product breakout data for wood from difference sources. This could lead to underestimation of the area of forest required to supply marginal HWP imports – particularly for Boreal (3) where the additional wood removed is not logs but lower-quality stumps and residues.

## Tropical afforestation

Afforestation is linked to increased timber demand[8], so whilst detrimental GWP impacts of increasing harvesting from existing systems are possible, increased demand could also trigger a beneficial GWP impact from afforestation (beyond temperate regions). Considerable opportunities in the tropics, due to large areas of underutilized land and high potential tree growth rates[73], mean tropical afforestation can make a meaningful contribution to IRW supply within the study period, although the high growth rate tree species matched to most tropical environments (such as *Eucalyptus* spp.) produce hardwood with predominantly different uses and markets than temperate or boreal softwood, but still with an important role in large-volume global wood markets. We, therefore estimate the possible impact of afforestation at a scale to deliver the average marginal import volume(s) over the study period. The modelled scenario involves land use change from tropical wet grassland[75] to eucalyptus plantation managed on a 10-year rotation[76] clear-fell harvest with mean annual increment (MAI) 35 $m^3$/ha/yr[75]. While reported lower MAI could be used[77,78], we made our selection as it provides a conservative estimate of the potential GWP impacts of tropical afforestation (i.e. land use change) since the area of forest required to supply demand is relatively low (given the high MAI and therefore high wood supply rate, and low C storage per ha) and because we account only for $CO_2$ sequestration into above-ground C stocks, not below-ground stocks.

We calculate the GWP (net $CO_2$e sequestration) impact of tropical afforestation in tonnes of $CO_2$e per $m^3$/yr of harvested wood and the associated land footprint to facilitate direct comparison with the range of intensified forest management scenarios described above. See Supplementary Data 3 for calculations.

## Marginal wood demand not met

The impact of marginal IRW demand not being met is calculated as a loss of the avoided emissions impact of the marginal imported HWP, i.e. consequential $CO_2e$ emissions from increased use of concrete and prolonged reliance on fossil fuels. It is calculated as an average impact per $1 m^3/yr$ over 100 years, to account for the effects of decarbonisation over the study period. See Supplementary Data 3 for calculations.

## Data availability

The data generated in this study are provided in the Supplementary Information and Source Data files. Background data were generated using the publicly-available CBM-CFS3 model and extracted from the Ecoinvent v.3.539 database. All subsequent calculations were undertaken using standard MS Excel functions. Source data are provided with this paper.

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

## Acknowledgements

This work was funded by the UK Natural Environment Research Council (NERC) Envision Doctoral Training Programme NE/L002604/1 (awarded to EF). Additional CASE studentship funding was provided (to EF) by Woodknowledge Wales (Ffarm Moelyci, Felin Hen Road, Tregarth, LL57 4BB, UK) and Coed Cymru (The Forest Hub, Unit 6, Dyfi Eco Park, Machynlleth, Powys, SY20 8AX, UK). We thank Göran Berndes for his valuable advice on the final framing of the study.

## Author contributions

E.F. led the original conception of the study, undertook data collection and analysis, and drafted the manuscript. D.S. and J.H. participated in the original conception of the study, informed study design and edited the manuscript.

## Competing interests

The authors declare no competing interests.
