## [Transparent Peer Review file · Nature Communications]

Temperate forests can deliver future wood demand and climate-change mitigation dependent on afforestation and circularity

Corresponding Author: Ms Eilidh Forster

Version 0:

Reviewer comments:

Reviewer #1

(Remarks to the Author)

The manuscript "Can temperate forests deliver both future wood demand and climate-change mitigation?" by Forster et al. presents the results of the lifecycle assessments of the global warming potential of wood supply under the growing wood demand scenarios. The analysis is fairly clear and thorough, however I have several reservations outlined below:

1. There are no uncertainty estimates around the wood supply and demand as well as GWP. Incorporating the uncertainty analysis into the study would provide insight into where the future research efforts should be directed in order to reduce the uncertainty, as well as help evaluate the confidence in the reported estimates. For an example of uncertainty analysis using CBM-CFS please refer to J.M. Metsaranta, C.H. Shaw, W.A. Kurz, C. Boisvenue, and S. Morken. 2017. Uncertainty of inventory-based estimates of the carbon dynamics of Canada's managed forest (1990–2014). *Canadian Journal of Forest Research*. 47(8): 1082-1094. <https://doi.org/10.1139/cjfr-2017-0088>

2. Although authors did state in the methods section that risks of pests disease, wind, fire, warming and other climate change effects would not significantly alter the study findings because the risks are highly uncertain and would apply similarly across the study, I don't think it is the case. Fire, drought, changing productivity due to climate change would not change the wood demand, however they could profoundly affect the wood supply. Given the length of the projections in the study, warming would substantially increase heterotrophic respiration, and therefore would alter GWP projections. Drought events could substantially and repeatedly reduce forest yield due via inhibition of the photosynthetic rate and increasing mortality rate. Warming temperatures and changing precipitation regimes are also likely to affect forest net primary production. Increasing fire frequency and severity could profoundly affect the wood supply. I don't think these effects are negligible for GWP and wood supply estimates, and therefore should be considered in the study, especially given the 100-year projection time.

3. The study results are generalized for a "temperate country", however most of the data used in the study are for the UK, why not make the study focused on the UK? It would make it easier to assess the feasibility of the proposed scenarios and allow to avoid generalized statements (as a reader, I had a little trouble with those).

Reviewer #2

(Remarks to the Author)

Summary: In this manuscript, authors present a combined application of forest carbon models and life cycle assessment (LCA) to estimate global warming (GW) impacts (+/-) of harvested wood products' (HWPs) value chain in a temperate country (appears to be UK). For projected high and low wood demands, the GW impacts are assessed for different scenarios within the country (changing rotation length of forests, increasing rate of production, and expansion of forest area) and overseas imports from non-temperate forests (tropical country). This kind of study is an interesting attempt to integrate two areas of the forest products value chain (forestry and HWPs) for policy implications. The results indicate that increased wood use is not a climate-change solution unless afforestation, increasing forest productivity under sustainable forest management, and mitigating demand increases through enhanced circularity and cascading of wood use are also integrated

into the strategy. I am offering below some comments/suggestions to improve the manuscript.

Major comments

103-104: The current production and consumption levels of this temperate country should have been characterized to visualize the gap between demand and supply for the reference year 2023, and the how different scenarios or intended decisions might close this gap and influence GW impacts.

107-108: What was the reason for selection of Sitka spruce forest in afforestation? The authors might discuss whether the results would be different if the forests were Douglas-fir or Western hemlock.

Figure 2: Please change y-axis units to Tg CO₂e.

234: More interpretations could be added in 'high wood demand projection' results.

264-267: Not clearly explained how GWP impact of alternative increased due to increase in overseas wood supply. Prolonged use of non-wood product and fuel alternatives?

278-279: Please add explanation to this sentence, it seems confusing that more imports from tropical afforestation is better.

295: Clarity is needed on what type of non-wood product is considered substituted by HWPs. Because substitution credits for two type of non-wood products can be different for same HWP.

340: BECCS is associated with 'energy substitution' like non-wood products avoided is associated with 'product substitution'. There aren't a lot of interpretations in the results and discussion that focused on energy substitution.

600: there is no 'ix' in the components of LCA system boundary. Including via ...?

724-740: This whole paragraph does not seem to be fit for methodology. Its more suitable for introduction.

747-748: What is the per capita timber consumption in UK, which translates to 30% increase in demand by 2050?

754: In this paragraph, it would be good if authors give a brief about the different HWP end-uses (primary and cascading uses) considered and maybe a justification for selecting HWP uses.

766: More clarity on defining the overseas forest type and carbon storage. Also, was the transportation distance and mode of transport included in the analysis?

785: Why 'GWP (forest C) impact...'? Forest carbon can be stored, emitted, or removed but cannot be equated directly to GWP.

Overall, the methods section appears weak to me and needs a thorough revision to ensure that work can be reproduced.

Minor comments

325 and 330: should it be trip or tip?

723: why question marks in middle of the sentence?

Double numbering in references

Some of the links in supplementary excel file are broken. Please check.

Reviewer #3

(Remarks to the Author)

The aim of this case study is to quantify the GHG mitigation potential of different measures or forest management options (in particular afforestation) in terms of meeting an increasing demand for wood, assuming that existing models for the forest sector tend to "underestimate" forest carbon fluxes.

In fact, in addition to the development of forest carbon stocks, the delayed release of biogenic carbon through the use of wood as material as well as potential shifting effects of the GHG emission balances associated with the life cycle of these wood-containing product systems and their potential product alternatives do also have an impact on the overall GHG balance.

However, the implementation of the presented approach to estimate the total GHG impact of different management scenarios compared to the defined reference, in our view appears to be completely inadequate in this study. It does not comply with applicable international standards and existing state-of-the-art knowledge.

While the modeling of carbon storage development in the forest using the internationally recognized Carbon Budget Model for the Canadian Forestry Service appears adequate, the methodological inadequacies relate in particular to the life cycle assessment methodology, on the basis of whose standard-compliant, consistent and transparent implementation avoided emissions through "product substitution" can be estimated in the first place. In order to adequately consider process chain emissions and based on those also "avoided emissions from product substitution", it is crucial to meet the internationally standardized requirements for life cycle assessment (including ISO 14040/44 and ISO 21930). The mere summation of unrelated LCA process information from a background database (here: Ecolvent) is inadequate - at least for the processes outside the forest along the processing and value chain. The data used for calculating "primary avoided emissions (FF/product substitution)" are also completely unsuitable.

Furthermore, the simplified calculation of assumed carbon storage effects through the use of wood as a material (harvested wood products) contradicts central core requirements in the calculation as set out in the methodological guidelines and requirements provided by IPCC (incl. e.g. the consideration of inherited emissions). In consequence, the HWP contribution through biogenic carbon storage as well as potential "avoided emissions" appear to be massively overestimated.

While it seems undisputed, even without the present study, that "the expansion of the (industrial) bioeconomy should be linked to the availability of raw materials in order to avoid unrealistic supply expectations," statements to the effect that "considerable HWP-C storage and product substitution credits can be achieved simultaneously" are not at all tenable on the basis of this simple and, in our view, completely methodologically inadequate implementation.

The simple comparison of a changing supply of forest wood with a modeled demand for this raw material (including potential gaps) can also be carried out without balancing all GHG emissions relevant to the forestry and wood sector.

Version 1:

Reviewer comments:

Reviewer #1

(Remarks to the Author)

The authors provided fairly comprehensive responses to the reviewers' comments. Yet, after reading the revised paper there is a dissonance between the level of detail in the manuscript and practical applicability of the findings. In the response to reviewers authors noted that this manuscript implements a framework laid out in an earlier paper published in Nature Communications, however the implementation is rather abstract, an interested party (e.g. government of a temperate country) would have to re-do the analyses (i.e. implement the framework to their specific country) and may get substantially different results. This makes me question the value of this particular manuscript, given lack of connection to any specific country, for which the feasibility of the generated estimates could be evaluated.

In my opinion, the treatment of the uncertainties associated with the effects of climate change and disturbances on productivity and GWP was fairly simplistic, not illustrated in all figures and not illustrated for GWP estimates.

Lastly, please correct the following: CBM-CFS stands for "Carbon Budget Model of the Canadian Forest Sector".

Reviewer #2

(Remarks to the Author)

Author Responses to Reviewer Comments

We are grateful to and thank the reviewers for taking time to consider our manuscript.

	Remarks to Author	Author Responses
	Reviewer #1:	
	The manuscript “Can temperate forests deliver both future wood demand and climate-change mitigation?” by Forster et al. presents the results of the lifecycle assessments of the global warming potential of wood supply under the growing wood demand scenarios. The analysis is fairly clear and thorough, however I have several reservations outlined below:	We thank the reviewer for the positive comments and valuable specific suggestions to further enhance the paper.
1	There are no uncertainty estimates around the wood supply and demand as well as GWP. Incorporating the uncertainty analysis into the study would provide insight into where the future research efforts should be directed in order to reduce the uncertainty, as well as help evaluate the confidence in the reported estimates. For an example of uncertainty analysis using CBM-CFS please refer to J.M. Metsaranta, C.H. Shaw, W.A. Kurz, C. Boisvenue, and S. Morken. 2017. Uncertainty of inventory-based estimates of the carbon dynamics of Canada’s managed forest (1990–2014). Canadian Journal of Forest Research. 47(8): 1082-1094. https://doi.org/10.1139/cjfr-2017-0088	The demand profiles modelled are based on the outcome of an extensive literature review of wood demand modelling studies. We recognise and account for the significant uncertainty surrounding future wood demand projections (indicated by the wide range of demand projections published in the literature) by modelling two different demand profiles in the present study. These two demand profiles capture the range in the published literature. LCA of forest management scenarios under each of these demand profile scenarios indicates the degree of uncertainty and sensitivity of results to future demand projection assumptions. In dealing with uncertainties for the supply side, we have taken the simple and clear approach of testing multiple scenarios of variation in future forest productivity (yield). We believe that this is an efficient way of integrating all of these causes of uncertainty listed by the reviewer 1 comment 2. We have also added the following text to the 'Discussion section' lines 375-379 : "There is an urgent need for more integrated evidence that incorporates holistic assessment of prospective forestry value chains alongside landscape dynamics (including forest management and expansion), at both national and global scales, including improved estimates of the potential impacts of climate change-linked threats to the future productivity of both temperate forests and the other forest biomes providing wood production." See continuation of this response in response to reviewer 1 comment 2 below.

	Remarks to Author	Author Responses
2	Although authors did state in the methods section that risks of pests disease, wind, fire, warming and other climate change effects would not significantly alter the study findings because the risks are highly uncertain and would apply similarly across the study, I don't think it is the case. Fire, drought, changing productivity due to climate change would not change the wood demand, however they could profoundly affect the wood supply. Given the length of the projections in the study, warming would substantially increase heterotrophic respiration, and therefore would alter GWP projections. Drought events could substantially and repeatedly reduce forest yield due via inhibition of the photosynthetic rate and increasing mortality rate. Warming temperatures and changing precipitation regimes are also likely to affect forest net primary production. Increasing fire frequency and severity could profoundly affect the wood supply. I don't think these effects are negligible for GWP and wood supply estimates, and therefore should be considered in the study, especially given the 100-year projection time.	As described in response to the previous comment, we believe that the approach to test multiple scenarios of variation in future forest productivity (yield) is an efficient way of integrating many of the causes of uncertainty listed by the reviewer. It is the approach that is most compatible with the LCA methodology of our study. Therefore, to address this specific reviewer requirement we have introduced a further pair of scenarios ('lower productivity' than the reference scenario) to model impact of possible ecosystem shocks (leading to yield reduction) that could arise from natural disturbance such as pests, disease, wind, drought and fire. These new scenarios are a modification of the 'reference rotation' existing forest (YC18, 50-yr rotation) + afforestation combination. The new scenarios are a transition of a) 15% and b) 30% of the existing forest from YC18 to YC12 ramping up over a 15 year period (remaining as a 50-yr harvest rotation), followed by recovery back to YC18 after one harvest rotation, ramping back down over a 15-yr period. (The afforestation assumptions were unchanged). Essentially the scenario represents a rapid shock of -two different intensities (affect on 15% & 30% of the total area), followed by recovery after harvest (at 50-yrs old). A detailed description of the basis for the parameterisation of these two scenarios (two and a half pages of text) based on a substantial new literature review (32 references cited) is provided in the new Supplementary Information file "Modelling of natural disturbances – 'reduced productivity' ". We recognise that natural disturbances, the intensity and rate of impact, and management responses to these are complex and therefore the modelling options available are vast. We believe the new scenarios selected correspond to contrasting realistic disturbance patterns presented in the reviewed literature and therefore offer additional insights to the other scenarios already modelled. We have included the new results in an updated version of 'Supplementary Data 3', alongside the results of all the other scenarios modelled in this paper. We modelled these two new scenarios under high and low demand projections and for the variation in afforestation rates and planting periods. We found that the new 'lower productivity' scenarios in this set led to higher net forest carbon loss (domestic and overseas) than the 'reference rotation' scenarios, as well as other changes, yet these disturbances do not change the overall results or findings of the study. We have added text on this to the methodology (lines 701-708) and results section of the manuscript

	Remarks to Author	Author Responses
		(lines 114-118; 169-172; and 201-204) and have included a more comprehensive description in the new Supplementary Information file (Supplementary Methods 1) in order to minimise additional word count.

	Remarks to Author	Author Responses
3	The study results are generalized for a “temperate country”, however most of the data used in the study are for the UK, why not make the study focused on the UK? It would make it easier to assess the feasibility of the proposed scenarios and allow to avoid generalized statements (as a reader, I had a little trouble with those).	We intentionally kept the analysis generic for a hypothetical temperate country rather than a UK-specific case study in order to provide evidence that has broad relevance. (The UK is not 'typical' in that it has a high baseline of imports and low domestic wood production.) This study is prospective in nature and focuses on the impacts of change, based on transparent assumptions for future changes (e.g. rate of decarbonisation, projected wood demand increase, shifts in forest management, different afforestation rates) that are not unique to the UK (or any other country) or dependant on historic conditions that may be unique to the UK. In order to perform the LCA we needed to make assumptions on typical product breakouts at the forest gate and at a sawmill and a representative wood product flow. Given limitations on public availability of this kind of data, we used data available for the UK as the basis for these assumptions. These assumptions are broadly representative across temperate countries beyond the UK given the similarities in technology and wood value chains. For example, typical rates of conversion of softwood to sawtimber at sawmills is similar across temperate regions. We have edited the Methodology 'Scope of LCA' section (e.g. lines 622-628) to try and improve the reconciliation between our use of UK data and generalised statements.

	Remarks to Author	Author Responses
	Reviewer #2	
	Summary: In this manuscript, authors present a combined application of forest carbon models and life cycle assessment (LCA) to estimate global warming (GW) impacts (+/-) of harvested wood products' (HWPs) value chain in a temperate country (appears to be UK). For projected high and low wood demands, the GW impacts are assessed for different scenarios within the country (changing rotation length of forests, increasing rate of production, and expansion of forest area) and overseas imports from non-temperate forests (tropical country). This kind of study is an interesting attempt to integrate two areas of the forest products value chain (forestry and HWPs) for policy implications. The results indicate that increased wood use is not a climate-change solution unless afforestation, increasing forest productivity under sustainable forest management, and mitigating demand increases through enhanced circularity and cascading of wood use are also integrated into the strategy. I am offering below some comments/suggestions to improve the manuscript.	We thank the reviewer for the positive comments and endorsement of the important findings of the paper.
	Major comments	
1	103-104: The current production and consumption levels of this temperate country should have been characterized to visualize the gap between demand and supply for the reference year 2023, and the how different scenarios or intended decisions might close this gap and influence GW impacts.	We intentionally defined a hypothetical scenario in which domestic production equals domestic consumption in the reference year 2023, as indicated in Figure 1. As described in our response to reviewer 1 comment 3 above, we have taken this hypothetical approach rather than characterising the actual production and consumption levels of a specific country in order to provide evidence that has broad relevance. (For example, the UK is not 'typical' in that it has a high baseline of imports and low domestic wood production.) It makes sense for a counterfactual lifecycle assessment (which is concerned with change from a baseline) to start the scenario with production = consumption rather than starting

	Remarks to Author	Author Responses
		with a deficit or surplus, which would make interpretation of results more challenging. We then visualise how the different scenarios close the gap between future (increasing) demand and supply in Figure 1 by colour coding the supply deficit in red (and the supply surplus in pink). We have also added some additional text in the Figure 1 title to improve this description.
2	107-108: What was the reason for selection of Sitka spruce forest in afforestation? The authors might discuss whether the results would be different if the forests were Douglas-fir or Western hemlock.	Sitka spruce was selected as it grows widely (native to west coast of Canada and the United States; now planted in 16 countries worldwide, including as the predominant plantation species in the UK, Ireland and Denmark - all countries with low forest cover). Sitka spruce also grows well in degraded upland sites where land is most economically viable for afforestation, unlike productive Douglas fir or western hemlock, which require lower altitude land that is of higher value for food production. LCA results would not be different if different species (or species mixes) were modelled unless a significantly different yield class (growth curve) was also assumed. It is the yield rather than the species choice that drives the GWP impact results (which of course would be related in specific geographic contexts but in a hypothetical forest study such as this, the yield class is specified independently by the modeller as a representative average across contexts). In this way, the choice of Sitka spruce is simply a proxy for commercial conifer species. However, we believe that the clarity of the manuscript benefits from naming the case species, which links to the UK sawmill and-representative wood product flow data used in the study.
3	Figure 2: Please change y-axis units to Tg CO₂e.	We have made this change.
4	234: More interpretations could be added in 'high wood demand projection' results.	We are pleased that Reviewer 2 sees further possible interpretations that we could include. We would welcome the opportunity to add more interpretations, however because of the journal's strict word limits this would necessitate making significant cuts to other parts of the text, which would sacrifice their clarity and content. Given the importance of these other components of the paper that would have to be cut, with regret we have concluded that adding more

	Remarks to Author	Author Responses
		interpretations to the 'high wood demand projection' results would be of net cost to the value of the paper.
5	264-267: Not clearly explained how GWP impact of alternative increased due to increase in overseas wood supply. Prolonged use of non-wood product and fuel alternatives?	We have added words in this sentence (line 290) to clarify that the GWP impact of the alternative is "curtailed HWP supply" i.e. prolonged use of non-wood product and fuel alternatives. We have also added further description to indicate the relative impact of the specific overseas scenarios ('Boreal 1,2&3, Tropical CVL and Tropical (afforestation)' in Fig. 4) in which emissions associated with supplying the wood shortfall are higher or lower than the emissions from 'curtailed HWP supply' (prolonged use of non-wood and fuel alternatives).
6	278-279: Please add explanation to this sentence, it seems confusing that more imports from tropical afforestation is better.	It is unclear to us why this is confusing as it seems to be clearly shown in the results reported in Fig. 4. We do acknowledge the potential disbenefits of new tropical afforestation in lines 363-364 ("numerous socio-economic⁵¹ and biodiversity conservation²⁰ caveats". In essence, increasing demand for fast-growing tropical tree species, which could be established on the large areas of degraded land in the tropics (if local socio-economic conditions make them available), could actually enhance terrestrial carbon stocks.
7	295: Clarity is needed on what type of non-wood product is considered substituted by HWPs. Because substitution credits for two type of non-wood products can be different for same HWP.	In response to this point, to avoid adding significant additional word count, we have added text to the methodology (line 616-617) and in the Fig. 5 title to direct readers to supplementary information for more detail on the substitution credits assumptions (supplementary figure 1 and supplementary Table 1). We also direct readers to Forster et al. (2021) where the methodology was originally published. We have also improved figure 5 in the present paper to convey, among other things, the product substitution assumptions more clearly.
8	340: BECCS is associated with 'energy substitution' like non-wood products avoided is associated with 'product substitution'. There aren't a lot of interpretations in the results and discussion that focused on energy substitution.	We have expanded the sentence referring to BECCS to mention product substitution and emphasise the permanent geological storage of biogenic carbon that contributes a carbon sink. Lines 368-369.

	Remarks to Author	Author Responses
9	600: there is no 'ix' in the components of LCA system boundary. Including via ...?	We have checked the numbering of components of the LCA system boundary in figure 5 and in the text and cannot find anything missing.
10	724-740: This whole paragraph does not seem to be fit for methodology. Its more suitable for introduction.	We believe that the paragraph is important for justifying the important selection of assumptions for the projected rate of demand increase used in the study, which is a core aspect of the study's methodology. We have edited and changed the order of the text to make the inclusion of this text in the methodology more fluent and justified. E.g. lines 779-785 have been moved from earlier in the section.
11	747-748: What is the per capita timber consumption in UK, which translates to 30% increase in demand by 2050?	We do not believe that including per capita timber consumption in the UK is relevant to the study. The 30% increase by 2050 statistic was included as this is a date that many published projections focus on, because it is an important date for many 'net zero' targets as mandated by the 2015 Paris Agreement of UNFCCC, and therefore a statistic that readers are likely to be interested in. As stated in our response to reviewer 2 comment 1, we have prioritised assumptions that are non-country-specific and forward looking wherever possible, to maximise the transferability of results. In essence, the critical (and transferable) mathematical/biophysical dynamic is the rate of demand increase from the baseline relative to the expansion rate of baseline forest area (through afforestation) and the (change in) productivity of existing (and new) forest.
12	754: In this paragraph, it would be good if authors give a brief about the different HWP end-uses (primary and cascading uses) considered and maybe a justification for selecting HWP uses.	The primary HWP end-uses are described in the methodology and further in the SI, including in a revised system diagram (Figure 5) indicating major processes and products. Details of cascading uses and also justification for the assumptions used are provided in Forster et al. (2021), which we cite in the methodology. We believe there is insufficient justification to repeat all of this detail here, especially on account of the word count restrictions.
13	766: More clarity on defining the overseas forest type and carbon storage. Also, was the transportation distance and mode of transport included in the analysis?	Description of the overseas forest types is provided in the following paragraphs (lines 827-856). We did not include assumed differences in transport distances between overseas and domestic HWPs because, in a previous study (Forster et al., 2023), we found that these differences made a very small contribution (less than 0.01%) relative to the net GWP impact of the value chain.

	Remarks to Author	Author Responses
14	785: Why 'GWP (forest C) impact...'? Forest carbon can be stored, emitted, or removed but cannot be equated directly to GWP.	GWP (forest C) impact is defined in the previous paragraph as the 'GWP impact of fluxes in forest carbon stocks'. We have added '(GWP (forest C))' after this definition to add clarity. It refers to the GWP impact associated with net CO ₂ e fluxes to/from the forest. Line 828 .
15	Overall, the methods section appears weak to me and needs a thorough revision to ensure that work can be reproduced.	We have carefully addressed each of the reviewers specific comments and assume that in doing so we have met the need expressed here.
	Minor comments	
16	325 and 330: should it be trip or tip?	Tip means to tilt, tumble or topple. In this context, 'the balance... can tip from sink to source', means the balance can tilt or shift from sink to source.
17	723: why question marks in middle of the sentence?	We have removed these.
18	Double numbering in references	We have removed these.
19	Some of the links in supplementary excel file are broken. Please check.	We have checked the supplementary Excel files and assume that the reviewer is referring to the 'HWP calculation module' (Supplementary Data 2). The broken links are the emissions factors cells that link to the original LCA worksheet (Supplementary Data 1) that generated them. These links do not work once the workbook is resaved in a different location. However, in the final version of these supplementary data files, which will be stored in a permanent repository or in the publisher's website, we will endeavour to ensure the links all work.

	Remarks to Author	Author Responses
	Reviewer #3 :	
1	The aim of this case study is to quantify the GHG mitigation potential of different measures or forest management options (in particular afforestation) in terms of meeting an increasing demand for wood, assuming that existing models for the forest sector tend to "underestimate" forest carbon fluxes.	

	Remarks to Author	Author Responses
2	In fact, in addition to the development of forest carbon stocks, the delayed release of biogenic carbon through the use of wood as material as well as potential shifting effects of the GHG emission balances associated with the life cycle of these wood-containing product systems and their potential product alternatives do also have an impact on the overall GHG balance.	We are not clear on whether this forms part of the reviewer's summary of the paper or is a comment requiring our response. In case it is the latter, we provide the following clarification. We agree with the reviewer about the importance of impacts throughout the wood products life cycle. Yet, the full climate mitigation effects of forestry are often under-represented because: (i) LCA approaches that (sometimes) better represent substitution and possible end-of-life carbon storage effects are usually only applied to specific products (partial wood flows out of forests); (ii) inventory approaches that attempt to represent all wood products, typically neglect short-lived wood products used for energy, and don't attribute substitution effects back to the forest sector; (iii) both LCA and inventory approaches often disregard cascading uses of wood, and associated second (and possible further) substiutions, along with extended carbon storage effects; (iv) only very prospective analyses consider future BECCS deployment that could lock up biogenic carbon indefinitely.
3	However, the implementation of the presented approach to estimate the total GHG impact of different management scenarios compared to the defined reference, in our view appears to be completely inadequate in this study. It does not comply with applicable international standards and existing state-of-the-art knowledge.	We are surprised by this comment, which we strongly believe does not reflect the content and rigour of our study or its compliance with the highest international standards. International standards apply to specific types of accounting, such as Environmental Product Declarations for (LCA of) wood products or UNFCCC guidelines for national GHG inventory accounting. In contrast, there are no such "standards" for the prospective consequential LCA of entire forest and multiple downstream product systems, which is the innovative methodological approach of our study. That is precisely what makes this approach so valuable - it transcends the accounting rules that are applied to deal with truncated system boundaries (e.g. how to allocate forest effects to specific downstream products). This paper builds on the state-of-the-art approach applied in our earlier, highly-cited manuscript published in Nature Communications (Forster et al., 2021), and applies it to a temperate forest system under different management and expansion regimes; this is highly novel. To reiterate, we are not attempting to calculate product footprints as per various international standards, but instead (as is befitting for a submission to Nature Communications) to provide a novel, holistic and rigorous analysis of the climate mitigation efficacy of different forest

	Remarks to Author	Author Responses
		management strategies, providing key new evidence for policy makers and an innovative novel approach for researchers.
4	While the modeling of carbon storage development in the forest using the internationally recognized Carbon Budget Model for the Canadian Forestry Service appears adequate, the methodological inadequacies relate in particular to the life cycle assessment methodology, on the basis of whose standard-compliant, consistent and transparent implementation avoided emissions through "product substitution" can be estimated in the first place. In order to adequately consider process chain emissions and based on those also "avoided emissions from product substitution", it is crucial to meet the internationally standardized requirements for life cycle assessment (including ISO 14040/44 and ISO 21930). The mere summation of unrelated LCA process information from a background database (here: EcoInvent) is inadequate - at least for the processes outside the forest along the processing and value chain. The data used for calculating "primary avoided emissions (FF/product substitution)" are also completely unsuitable.	As explained in our previous response, our study is categorically not an attributional LCA, which seems to have been the assumption of the reviewer. Instead it is a consequential LCA. That said, this study does fully comply with ISO 14040 and 14044 standards (which in reality are a basic framework to structure LCA), insofar as it: provides a clear goal and scope of the study; calculates an inventory that accounts for relevant changes within the expanded system boundary; applies GWP100 characterisation factors for the life cycle impact assessment phase; interprets results with clear regard for the question and methods used (now including important sensitivity analyses). Furthermore, this study uses system expansion in preference to allocation, as recommended in the allocation hierarchy advocated by the ISO standards. The study categorically does not sum "unrelated" LCA process information - incurred or avoided processes are clearly indicated in the system boundary (Supplementary Fig. 1 and in Fig. 5, also now improved), based on a consequential LCA approach that incorporates all changes associated with the scenarios vs the baseline counterfactual situation.

	Remarks to Author	Author Responses
5	Furthermore, the simplified calculation of assumed carbon storage effects through the use of wood as a material (harvested wood products) contradicts central core requirements in the calculation as set out in the methodological guidelines and requirements provided by IPCC (incl. e.g. the consideration of inherited emissions). In consequence, the HWP contribution through biogenic carbon storage as well as potential “avoided emissions” appear to be massively overestimated.	Our methodology does not contradict core requirements of the IPCC and from this comment we do not understand how the reviewer considers that it does so. The IPCC provide guidelines for a number of approaches to deal with HWP C storage at a national scale. All of them are based on the fact that biogenic carbon harvested from the forest is "lost" from the terrestrial system, but some of this carbon "reappears" in longer-lived HWPs (with a large deficit reflecting biomass assumed to be immediately oxidised via combustion for bioenergy or for kiln drying of wood in sawmills). Our approach fully respects the biogenic carbon balance - in fact much more explicitly than typical national inventory accounting (with which we are fully familiar), because flows into all main products, both short- and long-lived, are accounted for, and the duration of carbon storage in each of these products, and any subsequent products, is explicitly accounted for in our methodology. Again, this is the benefit of our expanded boundary approach (across products and through time) - it minimises the influence of "cut-off" rules and associated value judgements, which often profoundly influence carbon footprints at the wood product level.
6	While it seems undisputed, even without the present study, that "the expansion of the (industrial) bioeconomy should be linked to the availability of raw materials in order to avoid unrealistic supply expectations, " statements to the effect that "considerable HWP-C storage and product substitution credits can be achieved simultaneously" are not at all tenable on the basis of this simple and, in our view, completely methodologically inadequate implementation.	Without any further insight from the reviewer as to where our methodology is inadequate, it is difficult to address this comment. We acknowledge that our original system diagram did not fully represent the scope of our methodology (although it was described fully in the text and illustrated in Supplementary Fig.1). Therefore, we hope that the improved system diagram in the resubmitted manuscript helps understanding of precisely how we have linked forest C storage with HWP-C storage and substitution, and long-term CCS C-storage effects in a coherent manner. As previously mentioned, this builds on the methodology of Forster et al. (2021) published in Nature Communications.
7	The simple comparison of a changing supply of forest wood with a modeled demand for this raw material (including potential gaps) can also be carried out without balancing all GHG emissions relevant to the forestry and wood sector.	It is true that a simple comparison of changing supply of forest wood with demand could be carried out in isolation. However, that would not address the important policy question of what the climate mitigation effect would be of alternative forest strategies that aim to minimise the gap between future supply and demand. That is where this paper provides a unique and robust contribution to the scientific literature, for which its rigorous methodological consequential LCA approach is essential.

Author response to reviewer comments

We are once again grateful for the opportunity to respond to Reviewer comments. We have addressed the comments below and in doing so believe we have further enhanced the article, as outlined in our response. All changes to the manuscript are highlighted in 'tracked-changes'.

REVIEWER COMMENTS

Reviewer #1 (Remarks to the Author):

Comment 1

The authors provided fairly comprehensive responses to the reviewers' comments. Yet, after reading the revised paper there is a dissonance between the level of detail in the manuscript and practical applicability of the findings. In the response to reviewers authors noted that this manuscript implements a framework laid out in an earlier paper published in Nature Communications, however the implementation is rather abstract, an interested party (e.g. government of a temperate country) would have to re-do the analyses (i.e. implement the framework to their specific country) and may get substantially different results. This makes me question the value of this particular manuscript, given lack of connection to any specific country, for which the feasibility of the generated estimates could be evaluated.

Response to Comment 1

We thank Reviewer 1 for their further comments, to which we have given careful consideration. The main point raised here is that the methodology remains abstract, such that others would need to perform their own analysis. We fully agree that researchers and other interested parties should perform context-specific analysis, and not rely solely on the results of a generic country case study (of the kind we use here to demonstrate the rigour, power and applicability of the framework that we have developed in this paper). We have subtly revised framing to reflect this, with additional clarity and emphasis on the novel methodological framework we have developed. In a new SI, we outline the framework in diagram, table and text form, highlighting its substantial novel components not previously published, and describe its implementation with reference to important parameters and examples of datasets and models of the kind that are available to generate context-specific results. However, we maintain that using a generic temperate country with an even-aged forest as an illustrative case study provides important insights that do have broad relevance for readers, whereas using the context of a specific country would not, owing to the unique nature of its forest-age deviations. We have made small edits throughout the manuscript to further emphasise to readers the importance of tailoring studies to specific contexts, especially where the results of such studies are intended to inform policy.

The framework sets out our exploratory approach for identifying 'low regrets' climate solutions for the forestry value chain by evaluating a range of plausible future scenarios, using powerful integrated forest modelling and lifecycle assessment (LCA) (Supplementary Fig. 1 & Supplementary Methods 1).

The present study builds substantially on the detailed prospective dynamic LCA modelling of entire forest-wood value chains developed and explained in Forster et al. (2021). Crucially, in the present study the full downstream greenhouse gas (GHG) mitigation consequences of wood use and end-of-

life management are considered - including carbon storage and material and energy substitution within a decarbonising future economy. Novel aspects to the present study (shown in pink in the new Supplementary Fig. 1) include calculating potential future wood supply deficit by comparing projected wood demand curves to wood supply from a range of modelled (expanded) temperate forest management scenarios, and linking the supply deficit to marginal expansion of supply from other regions.

Elaboration of the framework within the SI will facilitate interested parties to follow our approach for a relevant study context (Supplementary Table 1). We signpost readers to this SI at multiple points in the manuscript.

Comment 2

In my opinion, the treatment of the uncertainties associated with the effects of climate change and disturbances on productivity and GWP was fairly simplistic, not illustrated in all figures and not illustrated for GWP estimates.

Response to Comment 2

We acknowledge the simple treatment of climate change and disturbances (which we note can vary considerably, geographically) for our generic temperate country study. However, we maintain that this approach is appropriate to illustrate the magnitude of effect associated with highly uncertain future events. Recent articles pertaining to climate mitigation modelling highlight the need for “robust” rather than optimised decision making in the face of deep uncertainty and high complexity (Workman et al. (2021) and Workman et al. (2024)). We now cite these articles to further justify our approach, and propose natural disturbance events as a key parameter to represent (where possible) this impact within the proposed framework when tailoring scenarios in future studies for specific context.

Comment 3

Lastly, please correct the following: CBM-CFS stands for "Carbon Budget Model of the Canadian Forest Sector".

Response to Comment 3

Corrections made.